# Effective radiative forcing from emissions of reactive gases and aerosols – a multi-model comparison

Gillian D. Thornhill[1], William J. Collins[1], Ryan J. Kramer[2], Dirk Olivié[3], Ragnhild B. Skeie[4,] Fiona M. O'Connor[5,] Nathan Luke Abraham[6,7,], Ramiro Checa-Garcia[8], Susanne E. Bauer[9], Makoto Deushi[10], Louisa K. Emmons[11], Piers M. Forster[12], Larry W. Horowitz[13], Ben Johnson[5], James Keeble[7], Jean-Francois Lamarque[11], Martine Michou[14], Michael J. Mills[11], Jane P. Mulcahy[5], Gunnar Myhre[4], Pierre Nabat[14], Vaishali Naik[13], Naga Oshima[10], Michael Schulz[3], Christopher J. Smith[12,18], Toshihiko Takemura[15], Simone Tilmes[11], Tongwen Wu[16], Guang Zeng[17], Jie Zhang[16].

[1]Department of Meteorology, University of Reading, Reading, RG6 6BB, UK

[2]Climate and Radiation Laboratory, NASA Goddard Space Flight Center, Greenbelt, MD 20771,USA, and Universities Space Research Association, 7178 Columbia Gateway Drive, Columbia, MD 21046, USA

[3]Norwegian Meteorological Institute, Oslo, Norway

[4] CICERO – Centre for International Climate and Environmental Research Oslo, Oslo, Norway

[5] Met Office, Exeter, UK

[6]National Centre for Atmospheric Science, U.K

[7]Department of Chemistry, University of Cambridge, Lensfield Road, Cambridge, CB2 1EW, U.K.,

[8]Laboratoire des Sciences du Climat et de l'Environnement, IPSL/CNRS, 91191 Gif Sur Yvette, France

[9] NASA Goddard Institute for Space Studies, USA

[10] Meteorological Research Institute, Tsukuba, Japan

[11] National Center for Atmospheric Research, Boulder, CO, USA

[12] University of Leeds, Leeds, UK

[13] NOAA, Geophysical Fluid Dynamics Laboratory (GFDL), Princeton, NJ 08540-6649

[14] CNRM, Université de Toulouse, Météo-France, CNRS, Toulouse, France

[15] Research Institute for Applied Mechanics, Kyushu University, Japan

[16] Climate System Modeling Division, Beijing Climate Center, Beijing, China

[17] National Institute of Water and Atmospheric Research (NIWA), Wellington, New Zealand

[18]International Institute for Applied Systems Analysis (IIASA), Laxenburg, Austria

*Correspondence to*: Gillian D. Thornhill (g.thornhill@reading.ac.uk)

**Abstract**

This paper quantifies the pre-industrial (1850) to present-day (2014) effective radiative forcing (ERF) of anthropogenic emissions of $NO_X$, VOCs (including CO), $SO_2$, $NH_3$, black carbon, organic carbon, and concentrations of methane, $N_2O$ and ozone-depleting halocarbons, using CMIP6 models. Concentration and

emission changes of reactive species can cause multiple changes in the composition of radiatively active species:
tropospheric ozone, stratospheric ozone, stratospheric water vapour, secondary inorganic and organic aerosol and
methane. Where possible we break down the ERFs from each emitted species into the contributions from the
composition changes. The ERFs are calculated for each of the models that participated in the AerChemMIP
experiments as part of the CMIP6 project, where the relevant model output was available.
The 1850 to 2014 multi-model mean ERFs ($\pm$ standard deviations) are -1.03 $\pm$ 0.37 Wm$^{-2}$ for SO$_2$ emissions, -
0.25 $\pm$ 0.09 Wm$^{-2}$ for organic carbon (OC), 0.15 $\pm$ 0.17 Wm$^{-2}$ for black carbon (BC) and for NH$_3$ it is -0.07 $\pm$
0.01Wm$^{-2}$. For the combined aerosols (in the piClim-aer experiment) it is -1.01 $\pm$0.25 Wm$^{-2}$. The multi-model
means for the reactive well-mixed greenhouse gases (including any effects on ozone and aerosol chemistry) are
0.67 $\pm$ 0.17 Wm$^{-2}$ for methane (CH$_4$), 0.26 $\pm$ 0.07 Wm$^{-2}$ for nitrous oxide (N$_2$O) and 0.12 $\pm$ 0.2 Wm$^{-2}$ for ozone-
depleting halocarbons (HC). Emissions of the ozone precursors nitrogen oxides (NO$_x$), volatile organic
compounds (VOC) and both together (O$_3$) lead to ERFs of 0.14 $\pm$ 0.13 Wm$^{-2}$, 0.09 $\pm$ 0.14 Wm$^{-2}$ and 0.20 $\pm$ 0.07
Wm$^{-2}$ respectively. The differences in ERFs calculated for the different models reflect differences in the
complexity of their aerosol and chemistry schemes, especially in the case of methane where tropospheric
chemistry captures increased forcing from ozone production.

**1.  Introduction**
The characterisation of the responses of the atmosphere, climate, and earth systems to various forcing agents is
essential for understanding, and countering, the impacts of climate change. As part of this effort there have been
several projects directed at using climate models from different groups around the world to produce a systematic
comparison of the simulations from these models, via the Coupled Model Intercomparison Project (CMIP), which
is now in its  6$^{th}$ iteration (Eyring et al., 2016). This CMIP work has been subdivided into different areas of interest
for addressing specific questions about climate change, such as the impact of aerosols and reactive greenhouse
gases, and the AerChemMIP (Collins et al., 2017) project is designed to examine the specific effects of these
factors on the climate. The aerosol and aerosol precursor species considered are sulphur dioxide (SO$_2$), black
carbon (BC), organic carbon (OC). The reactive greenhouse gases and ozone precursors are methane (CH$_4$),
nitrogen oxide (NO$_X$), volatile organic compounds (VOCs – including carbon monoxide),  nitrous oxide (N$_2$O)
and ozone-depleting halocarbons (HC).
The focus of this work is to characterise the effect of the change from pre-industrial (1850) to present day (2014)
in aerosols and their precursors, and chemically-reactive greenhouse gases (including species that affect ozone)
on the radiation budget of the planet, referred to as radiative forcing, as an initial step to understanding the response
of the atmosphere and earth system to changes in these components. In previous reports of the Intergovernmental
Panel on Climate Change (IPCC) the effect of the various forcing agents on the radiation balance has been
investigated in terms of the radiative forcing, (RF), which is a measure of how the radiative fluxes at the top of
atmosphere (TOA) change in response to changes in, e.g., concentrations or emissions of greenhouse gases and
aerosols. There have been several definitions of radiative forcing, (Forster et al., 2016;Sherwood et al., 2015),
which generally considered the instantaneous radiative forcing (IRF), or a combination of the IRF including the
adjustment of the stratospheric temperature to the driver, generally termed the stratospheric-temperature adjusted
radiative forcing. More recently (Boucher, 2013;Chung and Soden, 2015) there has been a move towards using

the effective radiative forcing (ERF) as the preferred metric, as this includes the rapid adjustments of the atmosphere to the perturbation, e.g. changes in cloud cover or type, water vapour, tropospheric temperature, which may affect the overall radiative balance of the atmosphere. In this work, ERF is calculated using two atmospheric model simulations both with the same prescribed sea-surface temperatures (SSTs) and sea ice, but one having the perturbation we are interested in investigating, e.g. a change in emissions or concentrations of aerosols or reactive gases. The difference in the net TOA flux between these two simulations is then defined as the ERF for that perturbation.

Previous efforts to understand the radiative forcing due to aerosols and reactive gases in CMIP simulations have resulted in a wide spread of values from the different climate models, in part due to a lack of suitable model simulations for extracting the ERF from, e.g., a specific change to an aerosol species. The experiments in the AerChemMIP project have been designed to address this in part, by defining consistent model set-ups to be used to calculate the ERFs, although the individual models will still have their own aerosol and chemistry modules, with varying levels of complexity and different approaches.

There are complexities in assessing how a particular forcing agent affects the climate system due to the interactions between some of the reactive gases; for example methane and ozone are linked in complex ways, and this increases the problem of understanding the specific contribution of each to the overall ERF when one of them is perturbed. An attempt to understand some of these interactions is discussed in Section 4.2 below.

The experimental set-up and models used are described in Section 2, the methods for calculating the ERFs for the aerosol and chemistry experiments are described in Section 3, and the results are discussed in section 4. Final conclusions are drawn in Section 5.

## 2. Experimental Setup

### 2.1 Models

This analysis is based on models participating in the Coupled Model Intercomparison Project (CMIP6) (Eyring et al., 2016), which oversees climate modelling efforts from a number of centres with a view to facilitating comparisons of the model results in a systematic framework. The overall CMIP6 project has a number of sub-projects, where those with interests in specific aspects of the climate can design and request specific experiments to be undertaken by the modelling groups. To understand the effects of aerosols and reactive gases on the climate, a set of experiments was devised under the auspices of AerChemMIP (Collins et al., 2017), described in Section 2.2.

The anthropogenic emissions of the aerosols, aerosol precursors and ozone precursors (excluding methane) for use in the models are given by Hoesly et al. (2018) and van Marle et al. (2017). Models use their own natural emissions (Eyring et al., 2016). The well-mixed greenhouse gases (WMGHG), $CO_2$, $CH_4$, $N_2O$ and halocarbons are specified as concentrations either at the surface or in the troposphere. Not all of the models include interactive aerosols, tropospheric chemistry and stratospheric chemistry, which is the ideal for the AerChemMIP experiments, but those models which do not include all these processes provide results for a subset of the experiments described in Section 2.2.

The models included in this analysis are summarised below, and in Table 1 with an overview of the model set-up, aerosol scheme and type of chemistry models used included. A more detailed description of each model and the aerosol and chemistry schemes used in each is available in the supplementary materials, Table S1.

The CNRM-ESM2-1 model (Séférian et al., 2019;Michou et al., 2020) includes an interactive tropospheric aerosol
scheme, and an interactive gaseous chemistry scheme only above the level of 560 hPa. The sulfate precursors
evolve to $SO_4$ using a simple dependence on latitude. The cloud droplet number concentration (CDNC) depends
on $SO_4$, organic matter and sea-salt concentrations, so the aerosol cloud-albedo effect is represented, although
other aerosol-cloud interactions are not.
The UKESM1 model (Sellar et al., 2020) includes an interactive stratosphere-troposphere gas-phase chemistry
scheme (Archibald et al., 2020) using the UK Chemistry and Aerosol (UKCA); (Morgenstern et al.,
2009;O'Connor et al., 2014) model. The UKCA aerosol scheme, called GLOMAP-mode is two-moment
simulation of tropospheric black carbon, organic carbon, $SO_4$ and sea salt. Dust is modelled independently using
the bin scheme of Woodward (2001). A full description and evaluation of the chemistry and aerosol schemes in
UKESM1 can be found in Archibald et al. (2020) and Mulcahy et al. (2020) respectively.
The MIROC6 model includes the Spectral Radiation-Transport Model for Aerosol Species (SPRINTARS) aerosol
model which predicts mass mixing ratios of the main tropospheric aerosols and models aerosol-cloud interactions
in which aerosols alter cloud microphysical properties and affect the radiation budget by acting as cloud
condensation and ice nuclei (Takemura et al., 2005;Watanabe et al., 2010;Takemura and Suzuki, 2019;Takemura,
2018;Tatebe et al., 2019).
The MRI-ESM2 model (Yukimoto et al., 2019) has the Model of Aerosol Species in the Global Atmosphere mark-
2 revision 4-climate (MASINGAR mk-2r4c) aerosol model, and a chemistry model, MRI-CCM2 (Deushi and
Shibata, 2011) which models chemistry processes for ozone and other trace gases from the surface to middle
atmosphere. The model includes aerosol-chemistry interactions, and aerosol-cloud interactions (Kawai et al.,
2019). The ERFs of anthropogenic gases and aerosols under present-day conditions relative to preindustrial
conditions estimated by MRI-ESM2 as part of the Radiative Forcing Model Intercomparison Project (RFMIP)
(Pincus et al., 2016) and AerChemMIP are summarized in Oshima et al. (2020).
The BCC-ESM1 model (Wu et al., 2019;Wu et al., 2020) models major aerosol species including gas-phase
chemical reactions, secondary aerosol formation, and aerosol-cloud interactions including indirect effects are
represented. It does not include stratospheric chemistry, so concentrations of ozone, $CH_4$, and $N_2O$ at the top two
model levels are the zonally and monthly values derived from the CMIP6 data package.
The NorESM2 model contains interactive aerosols and uses the OsloAero6 aerosol module (Seland et al., 2020),
(Olivié et al., in prep.) describes the formation and evolution of BC, OC, $SO_4$, dust, sea-salt and SOA. There is a
limited gas-phase chemistry describing the oxidation of the aerosol precursors DMS, $SO_2$, isoprene, and
monoterpenes and oxidant fields of OH, $HO_2$, $NO_3$ and ozone are prescribed climatological fields, and there is no
ozone chemistry in the model.
The GFDL-ESM4 model consists of the GFDL AM4.1 atmosphere component, (Dunne et al., 2020;Horowitz et
al., 2020) which includes an interactive tropospheric and stratospheric gas-phase and aerosol chemistry scheme.
Nitrate aerosols are explicitly treated in this model.
The CESM2-WACCM model includes interactive chemistry and aerosols for the troposphere, stratosphere and
lower thermosphere (Emmons et al., 2010); (Gettelman et al., 2019). The representation of secondary organic
aerosols follows the Volatility Basis Set approached (Tilmes et al., 2019).
The IPSLCM6A-LR-INCA (referred to subsequently as IPSL-INCA) model used for this analysis has interactive
aerosols but a limited gas-phase model. The aerosol scheme is based on a sectional approach with to represent the
size distribution of dust, sea- salt (which has an additional super-coarse mode to model largest emission of spray-
salt aerosols), BC, $NH_4$, $NO_3$, $SO_4$, $SO_2$ and OA with a combination of accumulation and coarse log-normal modes
with both soluble and insoluble treated as independent modes. DMS emissions are prescribed and not interactively
calculated. BC is modelled as internally mixed with sulphate (Wang et al. (2016), where the refractive index is
relies on Garnet-Maxwell method. Its emissions are derived from inventories. A new dust refractive index is
implemented (Di Biagio et al., 2019). Well mixed trace gases concentrations/emissions are forced with
AMIP/CMIP6 datasets (Lurton et al., 2020) ozone using Checa-Garcia et al. (2018) and solar forcing from Matthes
et al. (2017).
The GISS-E2-1 model aerosol scheme (One-Moment Aerosol (OMA)) module, which includes sulfate, nitrate,
ammonium, carbonaceous aerosols (BC and OC), is coupled to both the tropospheric and stratospheric chemistry
scheme. For the results reported here, the physics version 3 of this model configuration was used, which includes
the aerosol impacts on clouds. For details of the model, see Bauer et al. (2020).

**Table 1 Components used in the Earth system models (detailed Table is in Supplementary material, Table S1)**

|  | Aerosols | Tropospheric chemistry | Stratospheric chemistry |
|---|---|---|---|
| IPSL-CM6A-LR-INCA | Interactive | No | No |
| NorESM2-LM | Interactive | SOA and sulfate precursor chemistry | No |
| UKESM1-LL | Interactive Tropospheric. Prescribed stratospheric | Interactive | Interactive |
| CNRM-ESM2-1 | Interactive | Chemical reactions down to 560 hPa | Interactive |
| MRI-ESM2 | Interactive | Interactive | Interactive |
| MIROC6 | Interactive | SOA and sulfate precursor chemistry | No |
| BCC-ESM1 | Interactive | Interactive | No |
| GFDL-ESM4 | Interactive | Interactive | Interactive |
| CESM2-WACCM | Interactive | Interactive | Interactive |
| GISS-E2-1 | Interactive | Interactive | Interactive |

**2.2 Experiments**
The AerChemMIP timeslice experiments (Table 2) are used to determine the present-day (2014) ERFs for the
changes in emissions or concentrations of reactive gases, and aerosols or their precursors (Collins et al., 2017).
The ERFs are calculated by comparing the change in net TOA radiation fluxes between two runs with the same

prescribed sea surface temperatures (SSTs) and sea ice, but with near-term climate forcers (NTCFs - also referred to as short-lived climate forcers - SLCFs), reactive gas and aerosol emissions, and well-mixed greenhouse gases (WMGHG - methane, nitrous oxide, halocarbon) concentrations perturbed. It should be noted that in AerChemMIP the NTCF experiment excludes $CH_4$ the experimental design. The control run uses set 1850 pre-industrial values for the aerosol and aerosol precursors, $CH_4$ $N_2O$, ozone precursors and halocarbons, either as emissions or concentrations (Hoesly et al., 2018;van Marle et al., 2017;Meinshausen et al., 2017). Monthly varying prescribed SSTs and sea-ice are taken from the CMIP6 DECK coupled pre-industrial (1850) control simulation. Each experiment then perturbs the pre-industrial value by changing one (or more) of the species (emissions or concentrations) to the 2014 value, while keeping SSTs and sea-ice prescribed as in the pre-industrial control. Note adding individual species to a pre-industrial control will likely give different results to a setup where species were individually subtracted from a present-day control. The NTCFs are perturbed individually or in groups. This provides ERFs for the specific emission or concentration change, but also for all aerosol precursor or NTCFs combined (Collins et al., 2017). For models without interactive tropospheric chemistry "NTCF" and "aer" experiments are the same; in the case of NorESM2 for the NTCF experiments the model attempts to mimic the full chemistry by setting the oxidants and ozone to 2014 values. The WMGHG experiments include the effects on aerosol oxidation, tropospheric and stratospheric ozone, and stratospheric water vapour depending on the model complexity.

Thirty years of simulation are required to minimise internal variability (mainly from clouds) (Forster et al, 2016.), and one ensemble member was used for each experiment (almost all models provided only a single ensemble member).

**Table 2 List of fixed SST ERF simulations. (NTCF in (Collins et al., 2017) is also referred to as 'SLCF' - short-lived climate forcers - in other publications) and for the purposes of this study excludes methane.**

| Experiment ID | $CH_4$ | $N_2O$ | Aerosol Precursors | Ozone Precursors | CFC/ HCFC | Number of models |
|---|---|---|---|---|---|---|
| *piClim-control* | 1850 | 1850 | 1850 | 1850 | 1850 | 11 |
| *piClim-**NTCF*** | 1850 | 1850 | **2014** | **2014** | 1850 | 8 |
| *piClim-**aer*** | 1850 | 1850 | **2014** | 1850 | 1850 | 9 |
| *piClim-**BC*** | 1850 | 1850 | 1850 (non BC) **2014** (BC) | 1850 | 1850 | 7 |
| *piClim-**O3*** | 1850 | 1850 | 1850 | **2014** | 1850 | 4 |
| *piClim-**CH4*** | **2014** | 1850 | 1850 | 1850 | 1850 | 8 |
| *piClim-**N2O*** | 1850 | **2014** | 1850 | 1850 | 1850 | 5 |
| *piClim-**HC*** | 1850 | 1850 | 1850 | 1850 | **2014** | 6 |
| *piClim-**NOX*** | 1850 | 1850 | 1850 | 1850 (non $NO_x$) **2014** ($NO_x$) | 1850 | 5 |
| *piClim-**VOC*** | 1850 | 1850 | 1850 | 1850 (non CO/VOC) **2014** (CO/VOC) | 1850 | 5 |
| *piClim-**SO2*** | 1850 | 1850 | 1850 (non $SO_2$) **2014** ($SO_2$) | 1850 | 1850 | 6 |

| piClim-*OC* | 1850 | 1850 | 1850 (non OC) 2014 (OC) | 1850 | 1850 | 6 |
|---|---|---|---|---|---|---|
| piClim-*NH3* | 1850 | 1850 | 1850 (non NH$_3$) 2014 (NH$_3$) | 1850 | 1850 | 2 |

## 3. Methods

In the following analysis we use several methods to analyse the ERF and the relative contributions from different
aerosols, chemistry and processes to the overall ERF for the models and experiments described above, where the
appropriate model diagnostics were available.

### 3.1 Calculation of ERF using fixed SSTs

The ERF is calculated from the experiments described above, where the sea surface temperatures and sea-ice are
fixed to climatological values. Here the ERF is defined as the difference in the net TOA flux between the perturbed
experiments and the piClim-control experiment (Sherwood et al., 2015), calculated as the global mean for the 30
years of the experimental run (where the models were run longer than 30 years, only the last 30 years was used).
This allows us to calculate the ERF for the individual species based on the changes to the emission or
concentrations between the control and perturbed runs of the models. The assumption is that there is minimal
contribution from the climate feedback when the SSTs are fixed, but the resultant ERF includes rapid adjustments
to the forcing agent in the atmosphere (Forster et al., 2016).
The ERF calculated using this method includes any contributions to the ERF resulting from changes in the land
surface temperature ($T_s$), which ideally should be removed (Shine et al., 2003;Hansen et al., 2005;Vial et al., 2013)
(as the ocean temperature changes are removed by using fixed SSTs). However, there is no simple way to prescribe
land surface temperatures in the models considered here analogous to the fixing the SSTs, so we make the land
surface temperature correction by calculating the surface temperature adjustment from the radiative kernel (see
Section 3.2) and subtracting it from the standard ERF as calculated above (see also Smith et al. (2020a);(Tang et
al., 2019)). This is designated the ERF_ts to differentiate it from the standard ERF as described above.

### 3.2 Kernel Analysis

Where the relevant data are available, we use the radiative kernel method (Smith et al., 2018;Soden et al.,
2008;Chung and Soden, 2015) to break down the ERF into the instantaneous radiative forcing (IRF) and individual
rapid adjustments (designated by A) which are radiative responses to changes in atmospheric state variables that
are not coupled to surface warming. In this approach, ERF is defined as:
$$ERF = IRF + A_{t\_trop} + A_{t\_strat} + A_{ts} + A_q + A_a + A_c + e \qquad (1)$$
where $A_{t\_trop}$ is the troposphere temperature adjustment,  $A_{t\_strat}$ is the troposphere temperature adjustment, $A_{ts}$ is
the surface temperature adjustment, $A_q$ is the water vapour adjustment, $A_a$ is the albedo adjustment, $A_c$ is the cloud
adjustment, and e is the radiative kernel error. Individual rapid adjustments ($A_x$) are computed as:

$$A_x = \frac{\delta R}{\delta x} dx \qquad (2)$$

where $\frac{\delta R}{\delta x}$ is the radiative kernel, a diagnostic tool typically computed with an offline version of a GCM radiative transfer model that is initialized with climatological base state data and $dx$ is the climate response of atmospheric state variable $x$, diagnosed directly from each model. Cloud rapid adjustments ($A_C$) are estimated by diagnosing cloud radiative forcing from model flux diagnostics and correcting for cloud masking using the kernel-derived non-cloud adjustments and IRF, following common practice (e.g. (Soden et al., 2008;Smith et al., 2018)), whereby:

$$A_C = (ERF - ERF^{clr}) - (IRF\text{-}IRF^{clr}) - \sum_{x=[T,ts,q,a]} (A_x - A_x^{clr}) \qquad (3)$$

For the calculation of the IRF (for aerosols this is the direct effect) here, the clear-sky IRF ($IRF^{clr}$) is estimated as the difference between clear-sky ERF ($ERF^{clr}$) and the sum of kernel-derived clear-sky rapid adjustments ($A_x^{clr}$). Since estimates of $A_c$ are dependent on IRF, the same differencing method cannot be used to estimate IRF under all-sky conditions without special diagnostics (in particular the International Satellite Cloud Climatology Project diagnostics (ISCCP) diagnostics) not widely available in the AerChemMIP archive. Instead, for the calculations presented here all-sky IRF is computed by scaling $IRF^{clr}$ by a species-specific factor to account for cloud masking (Soden et al. 2008).

Kernels are available from several sources, and for this analysis we used kernels from CESM, (Pendergrass et al., 2018), GFDL (Soden et al., 2008), HadGEM3, (Smith et al., 2020b), and ECHAM6 (Block and Mauritsen, 2013) and took the mean from the four kernels for each model. Overall the individual kernels produced very similar results for each model, as reported in Smith et al. (2018).

**3.3 Calculation of ERF using aerosol-free radiative fluxes**

To understand the contributions of various processes to the overall ERF we can attempt to separate the ERF that is due to direct radiative forcing from that due to the effects of clouds. Greenhouse gases and aerosols can alter the thermal structure of the atmosphere and hence cloud thermodynamics (the semi-direct effect, (Ackerman et al., 2000)), and aerosols can act via microphysical effects (e.g. increasing the number of condensation nuclei and decreasing the effective radii of cloud droplets, referred to as the aerosol cloud albedo effect and the cloud lifetime effect (Twomey, 1974;Albrecht, 1989;Pincus and Baker, 1994). Following the method of Ghan (2013) the contribution of the aerosol-radiation interactions to the ERF can be distinguished from that of the aerosol-cloud interactions by using a 'double-call' method. This means that the model radiative flux diagnostics are calculated a second time but ignoring the scattering and absorption by the aerosol – referred to in the equations below with the subscript 'af'. The other effects of the aerosol on the atmosphere (i.e. cloud changes, stability changes, dynamics changes) will still be present, however. The IRFari as defined here is the direct radiative forcing from the aerosol, due to scattering and absorption of radiation. The cloud radiative forcing (ERFaci) due to the aerosol-cloud interactions is then obtained by using the difference between the aerosol-free all-sky fluxes and the aerosol-free clear-sky fluxes, which isolates the cloud effects (see Eqns. 4-6, where Eqn. (6) is included for completeness). The ERFaci may include non-cloud rapid adjustments in cloudy regions of the atmosphere. The final term is the ERF as calculated from fluxes with neither clouds nor aerosols (ERFcs,af).

The ERFs are calculated in the same way as for the all-sky ERF described in Section 3.1, except that the all-sky radiative flux diagnostics are replaced by the relevant aerosol-free fluxes for both the clear-sky and all-sky cases.


264   IRFari = (ERF – ERFaf)                   (4)

265   ERFaci = ERFaf – ERFcs,af                 (5)

266   ERFcs,af = ERFcs,af                   (6)

Separating the IRF in Eqn. (1) into aerosols and greenhouse gas contributions, IRF= $IRF_{aer}$+$IRF_{GHG}$, we can re-
write Eqns. 4-6.

269   $IRF_{ari}$ = $IRF_{aer}$                   (7)

270   $ERF_{aci} = A_C + \sum_{x=[T,ts,q,a]}(A_x - A_x^{clr}) + (IRF_{GHG} - IRF_{GHG}^{clr})$     (8)

271   ERFcs,af = $\sum_{x=[T,ts,q,a]} A_x^{clr} + IRF_{GHG}^{clr}$           (9)

So ERFaci is equivalent to $A_C$ in Eqn. (3) with extra terms to account for the all-sky - clear-sky difference in the
non-cloud adjustments and all-sky - clear-sky difference in any greenhouse gas IRF. With no greenhouse gas
changes ERFcs,af is the total clear-sky non-cloud adjustment. Ghan (2013) attributes this mostly to the surface
albedo change $A_\alpha^{clr}$, however the kernel analysis shows other non-cloud adjustments are larger (Table S4). For
greenhouse gases ERFcs,af is the total clear-sky ERF. Assuming the non-cloud adjustments are small apart from
$T_{strat}$ (Table S4), ERFcs,af is approximately $SARF_{GHG}^{clr}$. The $SARF_{GHG}^{clr}$ is expected to be an overestimate of
$SARF_{GHG}$ by 10-40% due to cloud masking (Myhre and Stordal 1997). Thus for greenhouse gases the ERFaci will
be a combination of the cloud adjustment and cloud-masking.
**4. Results**
**4.1 Aerosols and precursors**
**4.1.1 Inter-model Variability**
The ERFs are calculated as described in Section 3.1, and the summary chart of the ERFs is shown in Fig. 1 for
those models with available results – it should be noted that not all models ran all the experiments. The multimodel
mean is shown as a separate bar in Fig. 1, with the value given and the standard error indicated with error bars. A
table of the individual values for each model and the multimodel mean are included Table S2 in the supplementary
materials.

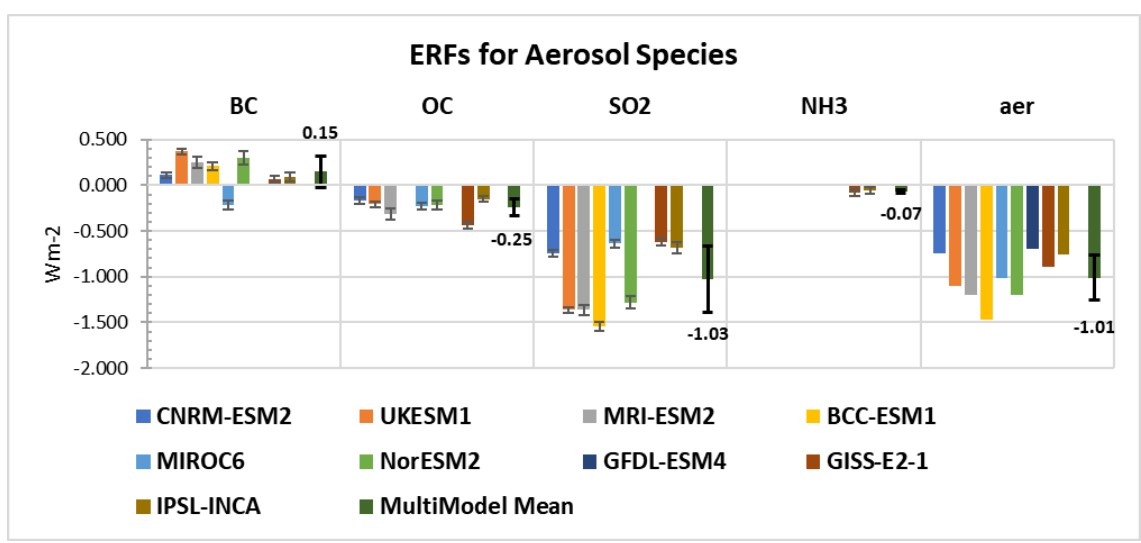

 **Fig. 1 Aerosol ERFs for the models with the available diagnostics for the aerosol species experiments, with interannual**
**variability represented by error bars showing the standard error. The piClim-aer experiments include the BC, OC**
**SO2 aerosols, and for GISS-E2-1 and IPSL-INCA NH3 aerosols are also included. The multimodel mean is shown with**
**the mean value and error bars indicating the standard deviation.**
For the piClim-BC results, the range of values is from -0.21 Wm$^{-2}$ to 0.37 Wm$^{-2}$, while the MIROC6 model has a
negative ERF for BC, contrasting with the positive values from the other models - see further discussion on this
in Section 4.1.2.
The experiments for the OC (organic carbon) have a range from -0.44 Wm$^{-2}$ to -0.15 Wm$^{-2}$, and the variability
between the models is much less than for the other experiments. The calculated ERFs for the SO2 experiment
show a variation from -1.54 Wm$^{-2}$ to -0.62 Wm$^{-2}$, with CNRM-ESM2-1,  MIROC6, IPSL-INCA and GISS-E2-1
at the lower end of the range.  These models show a smaller rapid adjustment to clouds which would account for
this (see fig S1); also note that CNRM-ESM2-1 does not include aerosol effects apart from the cloud-albedo
effect. The two models with results for the NH$_3$ (GISS-E2-1 and IPSL-INCA) experiment have ERFs of -0.08 and
-0.06 Wm$^{-2}$ respectively.
The piClim-aer experiment which uses the 2014 values of aerosol precursors and PI (pre-industrial) values for
CH$_4$, N$_2$O and ozone precursors shows a range from -1.47 Wm$^{-2}$ to -0.7 Wm$^{-2}$ among the models, making it
difficult to narrow the range of uncertainty of aerosols from global models. However, the range in the CMIP6
models is consistent with that reported in Bellouin et al. (2019), who suggest a probable range of -1.60 to -0.65
Wm$^{-2}$ for the total aerosol ERF, and compares well with the range of -1.37 to -0.63 Wm$^{-2}$ for the set of piClim-
aer experiments considered in (Smith et al., 2020a) as part of the RFMIP project. In general, the sum of the ERFs
from the individual BC, OC and SO$_2$ experiments does not equal the piClim-aer experiment, due to non-linearity
in the aerosol-cloud interactions, particularly since the aerosol perturbation is added to the relatively pristine pre-
industrial atmosphere. In the case of GISS and IPSL-INCA, and GFDl-ESM4 the models also include nitrate
aerosols.
The issue of the effect of perturbing the pre-industrial atmosphere with the aerosol changes is examined in more
detail in the Supplementary material (see section S6) for NorESM2, where a sensitivity analysis was carried out.
This analysis does not repeat the AerChemMIP experiments with the perturbation in a present-day atmosphere
but examines the effect of adding the SO$_2$ and combined aerosol perturbation to an already polluted present day
atmosphere. In this simplified sensitivity study the differences are 13% for the SO$_2$ experiment, and 20% for the
combined aerosol experiment. However, it should be borne in mind that this is for a specific model, and the
perturbed experiment still has the 1850 climate conditions.
The ERF_ts is a simplified method for corrections of land surface warming in fixed sea surface temperature
simulations which in addition to land surface changes leads to changes in land surface albedo changes,
tropospheric temperature, water vapor and cloud changes (Smith et al., 2020a;Tang et al., 2019).
The ERF_ts for the models where the land surface temperature adjustment is removed are also included in
Supplemental Tables S2 and S3, for comparison with the standard ERF. In general, the difference between the
two values is small, of the order of 5 -10%.

**4.1.2 Breakdown of the ERF into atmospheric adjustments and IRF**

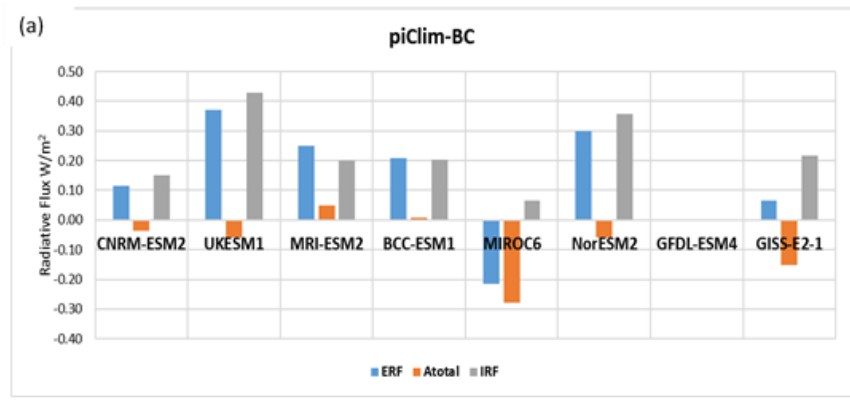

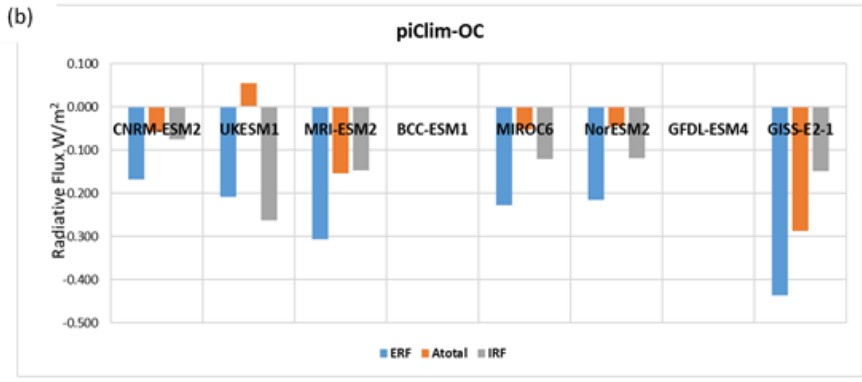

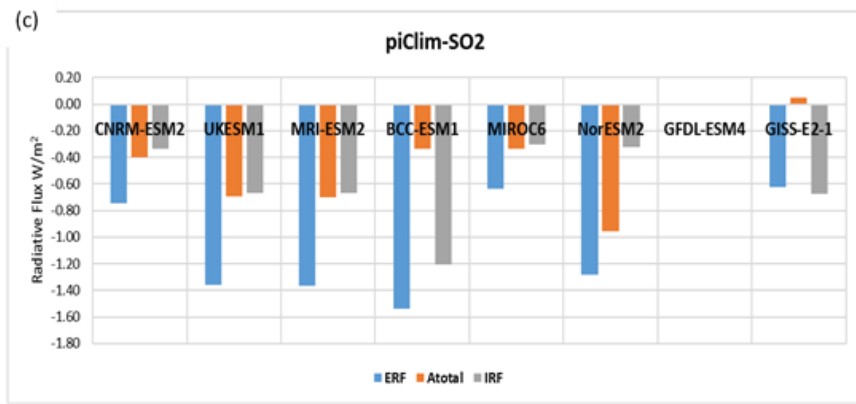

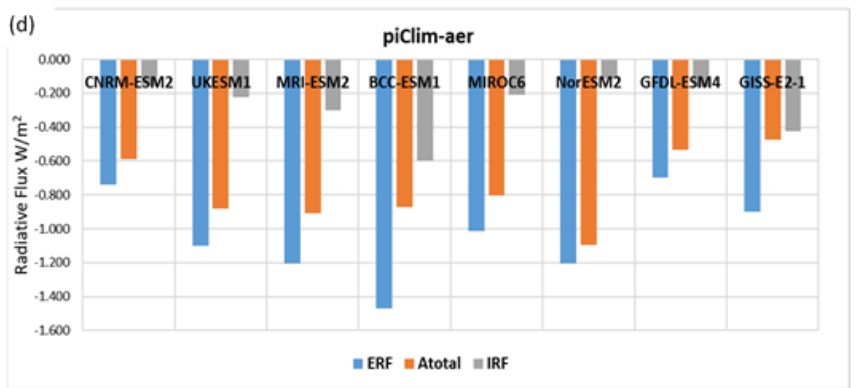

**Figure 2 Breakdown of the ERFs into the atmospheric rapid adjustments (Atotal) and IRF (instantaneous radiative forcing) for the aerosols. (a) piClim-BC experiment, (b) piClim-SO2 experiment, (c) piClim-OC experiment, (d) piClim-aer experiment**


The results in Fig. 2 show the ERF as calculated from the radiative fluxes in the fixed SST experiments (Section
3.1), the total of the atmospheric adjustments, $A_{total}$, described in Section 3.2 (where $A_{total} = A_T + A_{ts} + A_q + A_a +$
$A_c$ c.f. eqn. 1), and the instantaneous radiative forcing (IRF).
The sum of the IRF and the atmospheric adjustments should equal the overall ERF, however as the calculation of
the IRF depends upon an empirical factor for cloud masking to find the all-sky IRF from the clear-sky IRF (see
Section 3.2) the sum of the IRF and the $A_{total}$ will not necessarily equal the ERF as calculated directly from the
model radiative flux diagnostics. However, in general the difference is less than 3%, suggesting that the
approximation used in the calculation of the IRF is reasonable. Using the kernel method described above it is
important to note that the IRF calculated here accounts for the presence of the clouds but does not include cloud
changes such as the cloud albedo effect.
The models show a variability in the IRF for $SO_2$, (Fig. 2c) with a range of -0.3 $Wm^{-2}$ to -1.2 $Wm^{-2}$ with the BCC-
ESM1 model being the outlier, having the largest overall ERF. The OC experiments (Fig. 2b) range from -0.08
$Wm^{-2}$ to -0.26 $Wm^{-2}$, with a range for BC of 0.07 $Wm^{-2}$ to 0.43 $Wm^{-2}$ (Fig. 2a). In MIROC6 the treatment of BC
(Takemura & Suzuki 2019; Suzuki & Takemura 2019) leads to faster wet removal of BC and hence a lower IRF.
For the combined aerosols (Fig 2d) the range is from -0.1 $Wm^{-2}$ to -0.6 $Wm^{-2}$.
There are significant differences between the models in the $A_{total}$ for $SO_2$; these range from 0.05 $Wm^{-2}$ to -1.0 $Wm^{-}$
$^2$, where the differences are dominated by the cloud adjustments which here include the cloud albedo effect as part
of the adjustment (see Fig S3 for breakdowns of the atmospheric adjustments for all models). The adjustments to
BC are vary in sign and magnitude, with the MRI-ESM2 and BCC-ESM1 models having a slight positive
adjustment. The overall model mean has a weaker negative adjustment to that reported by (Stjern et al.,
2017;Samset et al., 2016;Smith et al., 2018). The MIROC6 model has a large negative adjustment which is large

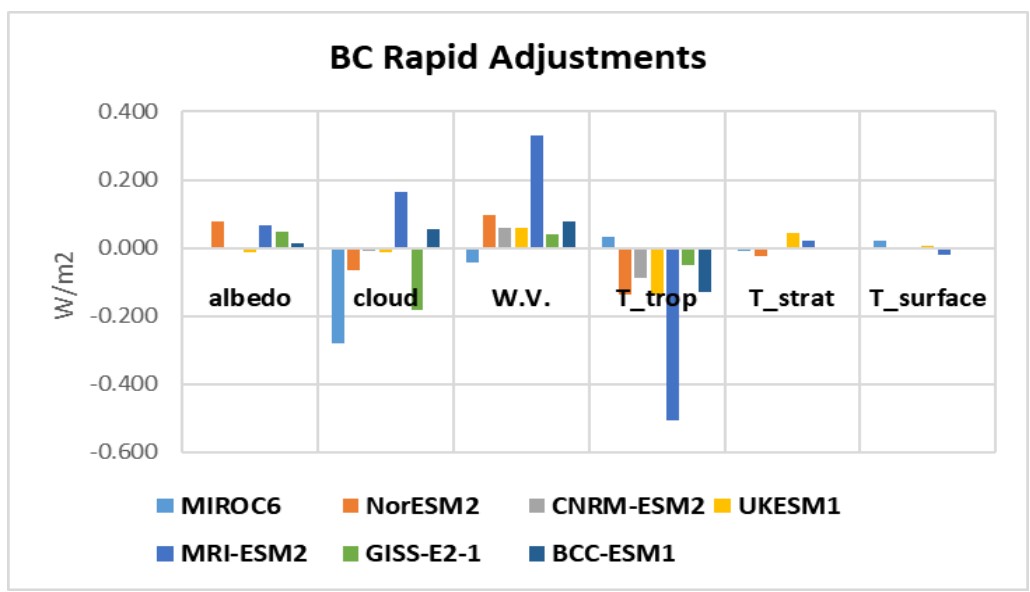

**Figure 3 Breakdown of the atmospheric adjustments (albedo, cloud, water vapour, troposphere temperature, stratosphere temperature and surface temperature) for the piClim-BC experiments, showing the variability between models.**

enough to lead to an overall negative ERF. We explore the contribution of the individual adjustments to BC in
more detail in Fig. 3.
Examining the breakdown of the rapid adjustments for the piClim-BC experiments (Fig. 3) we see considerable
variability in the relative importance of the rapid adjustments; the cloud adjustment dominates in MIROC6,
consistent with the increase in low clouds reported for this model, and the treatment of BC as ice nuclei causes
the large negative cloud adjustment here (Takemura and Suzuki, 2019;Suzuki and Takemura, 2019). The GISS-
E2-1 model also has a strong cloud rapid adjustment, but the larger positive value of the IRF leads to an overall
positive ERF for this model. With the exception of MIROC6 the negative tropospheric temperature adjustment is
balanced by the water vapour (specific humidity) adjustment, although the magnitude of these adjustments for
MRI-ESM2 is at least twice that for the other two models. The interaction of BC with clouds in the MRI-ESM2
model is discussed in detail in Oshima et al. (2020), in particular the impact of BC on ice nucleation in high
clouds. The larger surface albedo adjustment for both NorESM2 and MRI-ESM2 is most likely due to the
representation of deposition of BC on snow and ice in these models (Oshima et al., 2020).
The piClim-aer experiments (Fig. 1d) show all models have a negative $A_{total}$, covering a range from -0.47 to -1.1
$Wm^{-2}$. Overall, the cloud rapid adjustments dominate for the piClim-aer experiments, with a contribution ranging
from -0.45 to -1.1 $Wm^{-2}$ (See fig S1). Smith et al. (2020) also recently diagnosed forcing and adjustments in a
similar subset of CMIP6 models for the piClim-aer experiment as part of the Radiative Forcing Model
Intercomparison Project (RFMIP) efforts. While they also diagnosed IRF as a residual calculation between ERF
and the sum of rapid adjustments, they estimated cloud adjustments using a modified version of the APRP method
instead of radiative kernels. In their approach, the cloud albedo effect (i.e. Twomey Effect) is considered part of
the IRF, whereas in the traditional kernel decomposition, it is considered a cloud adjustment. Table S5 compares
the two sets of estimates, highlighting the IRF and total cloud adjustment exhibit a near equal absolute difference
between the two studies and the sum of IRF and total cloud adjustment are in close agreement (Mean % difference
~ 1.0% for this subset of models). This indicates the classification of the first indirect effect is the only noticeable
difference between the two approaches.
The breakdown of the rapid adjustments for all the models are included in supplementary Figure S1, showing the
contributions from each type of rapid adjustment for all the experiments for which we have the relevant
diagnostics.

**4.1.3 Radiation and Cloud interactions**

The second method of breaking down the ERF to constituents is described in Section 3.3, (the Ghan method), the
results from which are shown in Table 3. The detailed ERF results for MRI-ESM2 are summarized in Oshima et
al. (2020), and for UKESM1 in O'Connor et al. (2020a) . Only four of the models under consideration have so far
produced the necessary diagnostics for this calculation, and the results are presented in Table 3. For the
experiments on aerosols (aer, BC, $SO_2$, OC) the ERFcs,af (non-cloud adjustments) contribution is small, and the
ERF is largely a combination of the direct radiative effect IRFari, and the cloud radiative effect, ERFaci. The
IRFari is the direct effect of the aerosol due to scattering and absorption, while the ERFaci is the contribution
from the aerosol-cloud interactions and is approximately equal to the rapid adjustments due to clouds (*Ac* see
Section 3.2).



**Table 3 Results for IRFari, ERFaci and ERFcs,af  for aerosol experiments from several models**

| | UKESM1 | | | CNRM-ESM2 | | | NorESM2 | | | MRI-ESM2 | | |
|---|---|---|---|---|---|---|---|---|---|---|---|---|
| | IRFari | ERFcs,af | ERFaci | IRFari | ERFcs,af | ERFaci | IRFari | ERFcs,af | ERFaci | IRFari | ERFcs,af | ERFaci |
| aer | -0.15 | 0.05 | -1.00 | -0.21 | 0.08 | -0.61 | 0.03 | -0.03 | -1.21 | -0.32 | 0.09 | -0.98 |
| BC | 0.37 | 0.001 | -0.005 | 0.13 | 0.01 | -0.03 | 0.35 | 0.07 | -0.12 | 0.26 | 0.08 | -0.09 |
| OC | -0.15 | -0.01 | -0.07 | -0.07 | 0.04 | -0.14 | -0.07 | 0.02 | -0.16 | -0.07 | -0.05 | -0.21 |
| SO2 | -0.49 | 0.03 | -0.91 | -0.29 | 0.08 | -0.53 | -0.19 | -0.09 | -1.01 | -0.48 | 0.05 | -0.93 |


For the BC experiment the contribution of the aerosol-cloud interaction has a strong contribution to the overall
ERF, except in the case of UKESM1 where it is much weaker; this may be due to the strong SW and LW cloud
adjustments in this model cancelling out (O'Connor et al., 2020;Johnson et al., 2019). The $SO_2$ experiment shows
a large cloud radiative effect, in fact the ERFaci is mostly double the IRFari in all the models, due to the large
effect on clouds of $SO_2$ and sulfates through the indirect effects. For the OC experiments the ERFaci to IRFari
comparison is mixed, with the ERFaci general half or less the IRFari, except in the case of UKESM1, where this
ratio is reversed.
The IRFari are compared with the IRF calculated via the kernel analysis (Section 3.2) where the relevant model
results are available. These are shown in fig S2(a), the agreement is generally good giving confidence in the kernel
analysis. Similarly ERFaci compares well with the cloud adjustment Ac (fig S2(b)).

**4.1.4 AOD Forcing Efficiencies**
In order to break down the contributions of the constituent aerosol species to the overall aerosol ERF, we use the
AOD (aerosol optical depth) as a forcing efficiency metric for each of the species, and use this to assess their
contributions to the overall ERF. Not all models had diagnostics available for the AOD for the individual species,
so the analysis uses a subset of the models.
By looking at the single species piClim-BC, piClim-OC and piClim-SO2 experiments we can find the change in
the AOD for the individual species (e.g. ΔAOD for BC for the piClim-BC experiment), and use this to scale the
piClim-BC ERF by the AOD change. This assumes that the ERF in the single-species experiment is wholly due
to the change in that species as indicated by the AOD, an assumption which is explored in the Supplementary
material in Section S4. Table 5 shows the AOD forcing efficiency for the piClim-BC, piClim-SO2 and piClim-
OC experiments for each of the five models which had the relevant optical depth diagnostics available.

 **Table 4 Values of ERF, ΔAOD and ERF/AOD for aerosol experiments for CNRM-ESM2-, MIROC6, Nor-ESM2, GISS-**

 **E2-1 and MRI-ESM2  models.**

| BC Exp | BC ERF | Change in BC AOD | ERF/AOD |
|---|---|---|---|
| CNRM-ESM2 | 0.114 | 0.0015 | 77.64 |
| MIROC6 | -0.214 | 0.0006 | -339.38 |
| NorESM2 | 0.300 | 0.0019 | 159.75 |
| GISS-E2-1 | 0.065 | 0.002 | 31.65 |
| MRI-ESM2 | 0.251 | 0.0073 | 34.22 |
|  |  |  |  |
| OC Exp | OC ERF | Change in OA AOD | ERF/AOD |
| CNRM-ESM2 | -0.169 | 0.0030 | -57.20 |
| MIROC6 | -0.227 | 0.0065 | -35.05 |
| NorESM2 | -0.215 | 0.0053 | -40.57 |
| GISS-E2-1 | -0.438 | 0.0041 | -107.16 |
| MRI-ESM2 | -0.317 | 0.0034 | -94.39 |
|  |  |  |  |
| SO2 Exp | SO2 ERF | Change in SO4 AOD | ERF/AOD |
| CNRM-ESM2 | -0.746 | 0.0118 | -63.22 |
| MIROC6 | -0.637 | 0.0152 | -41.91 |
| NorESM2 | -1.281 | 0.0099 | -129.24 |
| GISS-E2-1 | -0.622 | 0.0308 | -20.22 |
| MRI-ESM2 | -1.365 | 0.0279 | -49.08 |

The MIROC6 model results in a negative scaling for BC due to the negative ERF for this experiment for this
model (Takemura & Suzuki 2019; Suzuki & Takemura 2019)  (see Section 4.1.1). The change in the BC AOD is
similar for CNRM-ESM2-1 and Nor-ESM2, and the scale factors reflect the differences in the ERF. The scaling
for the SO4 in the NorESM2 experiment is twice that of the other models, suggesting a larger impact of the SO4
AOD on the ERF in this model. These values differ somewhat from those found in Myhre et al. (2013b) where
they examined the radiative forcing normalised to the AOD using models in the AeroCom Phase II experiments.
They found values for sulfate ranging from -8 Wm$^{-2}$ to -21 Wm$^{-2}$ per unit AOD, much weaker than those in our
results. However, it is important to note that in the AeroCom Phase II experiments the cloud and cloud optical
properties are identical between their control and perturbed runs, so no aerosol indirect effects are included, nor

is any rapid adjustments (IRFari in Eqn. 4). For the BC experiment their values range from 84 Wm$^{-2}$ to 216 Wm$^{-2}$ per unit AOD, broadly similar to the results presented here (with the exception of the negative MIROC6 result). Their results for OA (organic aerosols) which include fossil fuel and biofuel emissions have values ranging from -10 Wm$^{-2}$ to -26 Wm$^{-2}$ per unit AOD, weaker than our values for the piClim-OC experiments which range from -35 Wm$^{-2}$ to -107 Wm$^{-2}$ per unit AOD but include the cloud indirect effects here.

The sum of the individual AODs from BC, SO$_4$, OA, dust and sea salt gives the total aerosol AOD in the piClim-aer experiment, where the various aerosols were combined. We can then use the AOD for each aerosol in the piClim-aer experiment and the forcing efficiency above to find the contribution of the individual aerosol to the overall change in ERF, providing an approximate estimate of the relative contribution of each aerosol to the overall ERF. In Fig. 4 the relative contributions to the ERF from black carbon (BC), organic aerosols (OA) and sulfate (SO$_4$) are shown for three of the models. The sum of the ERFs from the individual species is also compared to the ERF calculated from the piClim-aer experiment (NB the sea salt and dust contributions to the ERF are less than

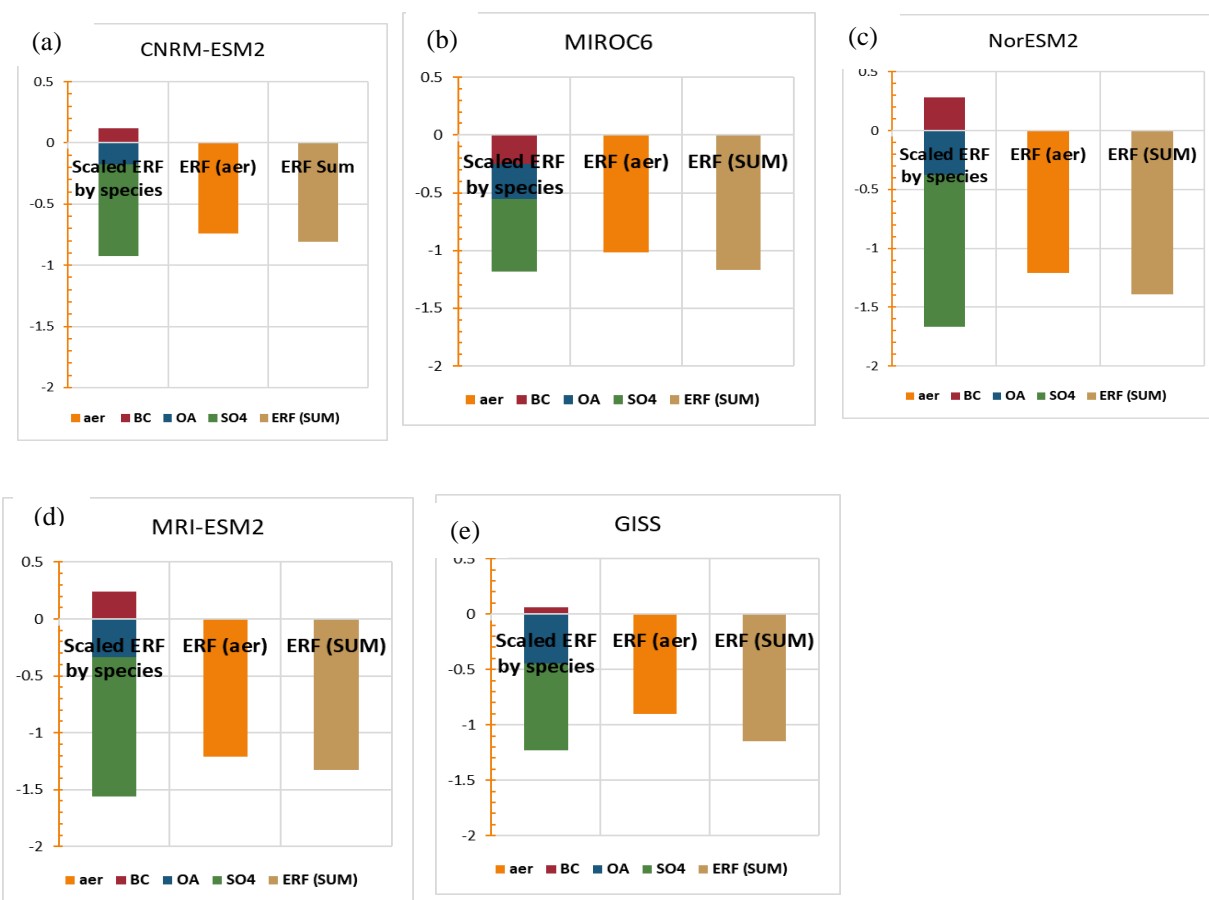

**Fig. 4 The contributions to the ERF for piClim-aer from the individual species, the sum of the scaled ERFs and the ERF calculated directly from the piClim-aer experiment for five of the models.**

1%, and not shown in this figure for clarity - the ERF/AOD forcing efficiency for these is presented in (Thornhill et al., 2020). There is considerable variation in the ERF for the piClim-aer experiments between models (see Section 4.1), but from this analysis the SO$_4$ is the largest contributor in all cases, although in the case of the MIROC6 model its relative importance is reduced. The positive ERF contribution from the BC tends to partly

offset the negative ERF from the OA and SO$_4$, except in the MIROC6 model, where the BC has a negative
contribution to the ERF.
The difference between the calculated ERF from the sum of the scaled ERFs is a result of the non-linearity of the
aerosol-cloud interactions, a factor which is increased because the aerosols are added to the pre-industrial
atmosphere. However, using the IRFari instead of the total ERF to calculate the forcing efficiency and using the
same method also results in a difference between the total IRFari derived from the scaled individual experiments
and the IRFari for the combined aerosol experiment, suggesting that the difference is not simply a result of the
aerosol-cloud interactions.
Using the burden as a scaling factor following the same analysis as described for the AOD results in a largely
similar result for the scaling factor, although interestingly the burden scaling for SO2 in the Nor-ESM2 model is
similar to the other models (see Table S6 for the burden forcing efficiency).

**4.2 Reactive greenhouse gases**
The different Earth system models include different degrees of complexity in their chemistry, so their responses
to changes in reactive gas concentrations or emissions differ. NorESM2 has no atmospheric chemistry, so there is
no change to ozone (tropospheric or stratosphere) or to aerosol oxidation following changes in methane or N$_2$O
concentrations. CNRM-ESM2-1 includes stratospheric ozone chemistry, but no non-methane hydrocarbon
chemistry and so ozone is prescribed below 560 hPa. There are no effects of chemistry on aerosol oxidation. BCC-
ESM1 includes tropospheric chemistry, but not stratospheric chemistry. Stratospheric concentrations are relaxed
towards climatological values. UKESM1, GFDL-ESM4, CESM2-WACCM, GISS-E2 and MRI-ESM2 all include
tropospheric and stratospheric ozone chemistry as well as changes to aerosol oxidation rates. The ERFs calculated
for the reactive gases for several models are shown in Fig. 5, with the multi-model means given in Supplementary
Table S3.
The contributions from gas-phase and aerosol changes to the ERF can be pulled apart to some extent by using the
clear-sky and aerosol-free radiation diagnostics (Table 5). The direct aerosol forcing (IRFari) is diagnosed as for
the aerosol experiments (section 3.3). The diagnosed changes in aerosol mass are shown in Table S8. GFDL-
ESM4 and GISS-ES-1 include nitrate aerosol and show expected responses from NO$_X$ emissions (including O3
experiment). CESM2-WACCM shows an increase in secondary organic aerosol from VOC emissions. Sulphate
responses are generally inconsistent across the models. There seems little correlation between aerosol mass
changes and diagnosed IRFari.
For gas-phase experiments the diagnosed cloud interactions (ERFaf-ERFcs,af) comprise the ERFaci from effects
on aerosol chemistry (as in section 3.3) but also any cloud adjustments and effects of cloud masking on the gas-
phase forcing (Eqn. 8). The clear-sky aerosol-free diagnostic (ERFcs,af) is an indication of the greenhouse gas
forcing however this will be an over-estimate as it neglects cloud masking effects (section 3.3).

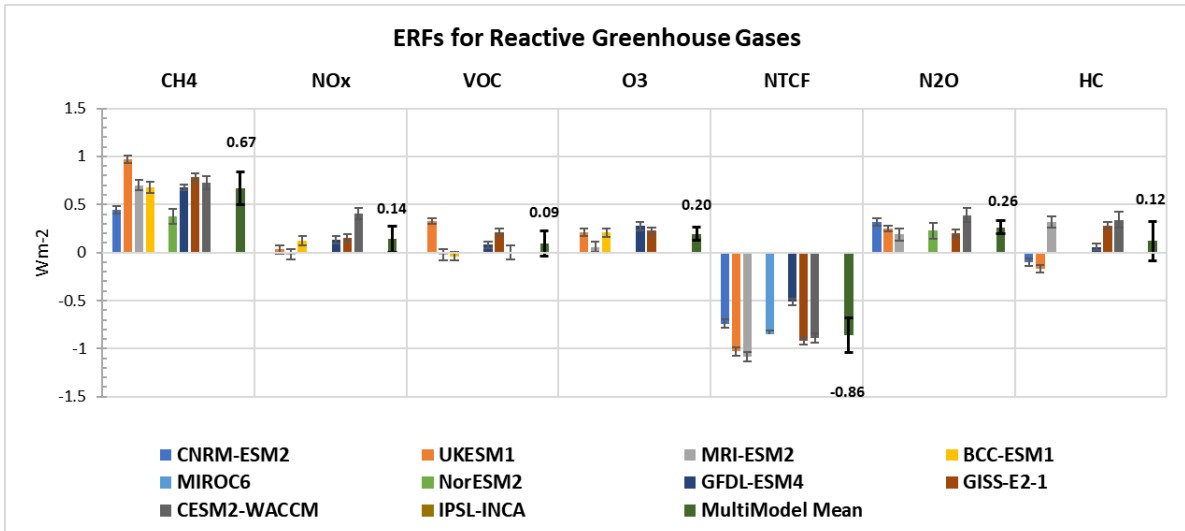

**Fig. 5 Reactive gas ERFs for the models with the available diagnostics for the reactive gas experiments with interannual variability represented by error bars showing the standard error. The multimodel mean is shown with the mean value and error bars indicating the standard deviation.**

### 4.2.1 ERF vs SARF

For the reactive greenhouse gases the kernel analysis is used to break down the ERF into the stratospherically adjusted radiative forcing (SARF), which is calculated using the IRF from the kernel analysis (Section 3.2) and the stratospheric temperature adjustment ($A_{t\_strat}$) ($SARF = IRF + A_{t\_strat}$), and the tropospheric adjustments, $A_{trop}$, which is the sum of the tropospheric atmospheric adjustments. These quantities are plotted in Fig. 6.

For methane the ERFs are largest for those models that include tropospheric ozone chemistry reflecting the increased forcing from ozone production, see section 4.2.2. The analytic calculation for $CH_4$-only based on Etminan et al. (2016) gives a SARF of 0.56 $Wm^{-2}$. The tropospheric adjustments are negative for all models except UKESM1 (Fig 6). The negative cloud adjustment comes from an increase in the LW emissions, possibly due to less high cloud. In UKESM1 (O'Connor et al., 2020b) show that methane decreases sulfate new particle formation, thus reducing cloud albedo and hence a positive cloud adjustment in that model.

For $N_2O$ results are available for models CNRM-ESM2, NorESM2, MRI-ESM2, and GISS-E2 (the analytic $N_2O$-only calculation gives a SARF of 0.17 $Wm^{-2}$). There appears little net rapid adjustment to $N_2O$ apart from CESM2-WACCM. Note that due to the method of calculating the all-sky IRF (section 3.2), the IRF and the adjustment terms do not sum to give the ERF.

The models respond very differently to changes in halocarbons. The expected halocarbon-only SARF is +0.30 $Wm^{-2}$ depending on exact speciation used in the model (WMO 2018). For CNRM-ESM2, UKESM1 and GFDL-ESM4 the ERFs are negative or only slightly positive (see also Morgenstern et al. (2020)), whereas for GISS-E2-1 and MRI-ESM2 the ERFs and SARF are both strongly positive. The differences in stratospheric ozone destruction in these models can partially explain the inter-model differences (section 4.2.2).


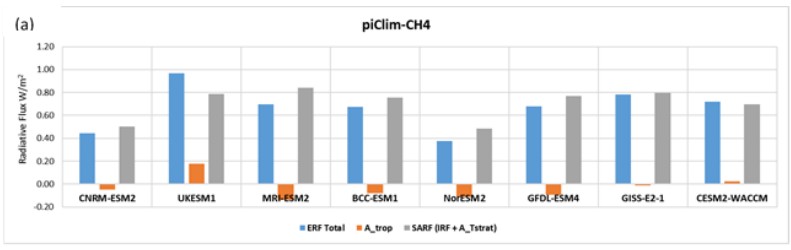

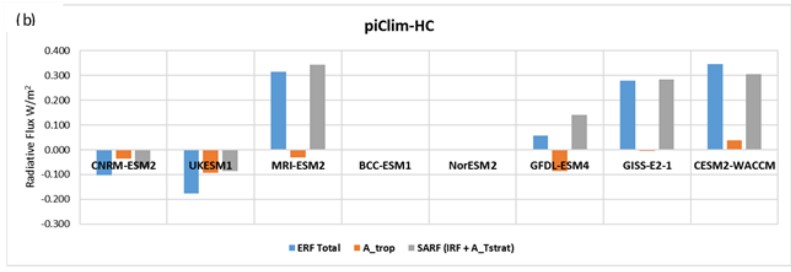

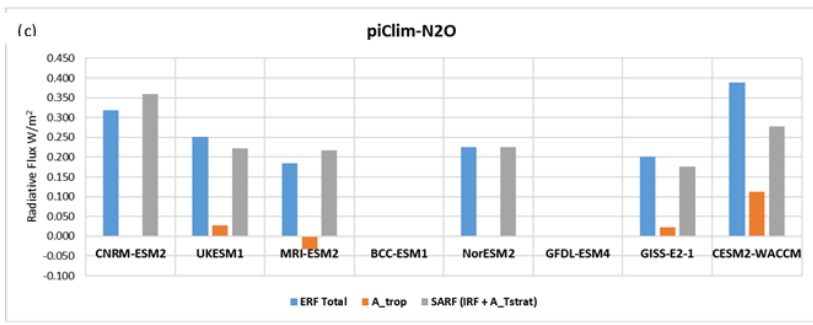

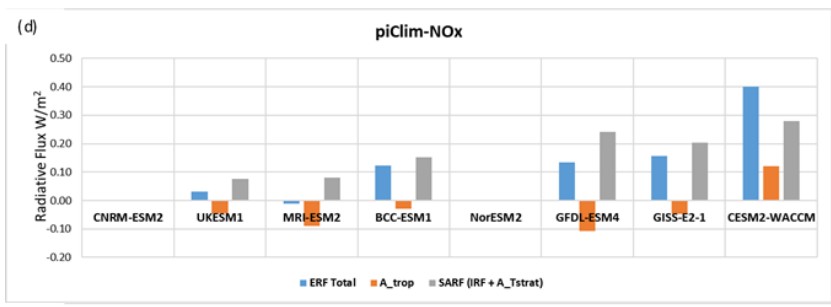

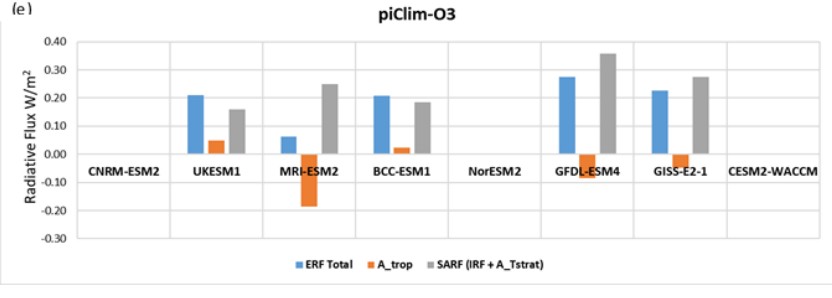

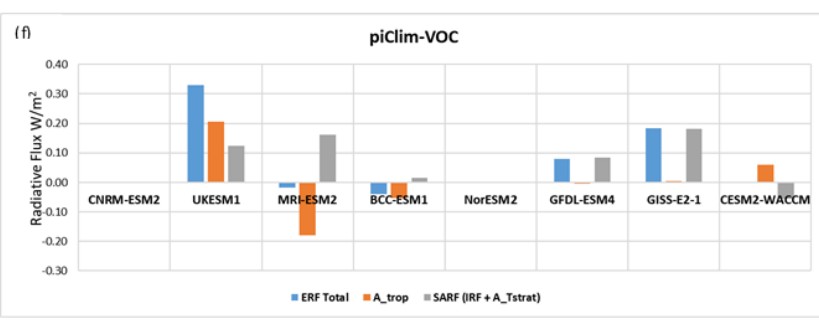


**Figure 6 Breakdown of the ERF into SARF  (IRF + A $_{t\_strat}$)and tropospheric rapid adjustments (A $_{trop}$) for the chemically reactive species (a) for piClim-CH4 experiments, (b) for piClim-HC experiments, (c) for piClim-N2O experiments, (d) for piClim-NOx experiments, (e) for piClim-O3 experiments, and (f) for piClim-VOC experiments**



**Table 5 Calculations of IRFari, ERFaci (cloud) and ERFcs,af for the chemically reactive species**

| | UKESM | | | GFDL-ESM4 | | | CNRM-ESM2 | | | NorESM2 | | | MRI-ESM2 | | |
|---|---|---|---|---|---|---|---|---|---|---|---|---|---|---|---|
| | IRFari | ERFcs,af | cloud | IRFari | ERFcs,af | cloud | IRFari | ERFcs,af | cloud | IRFari | ERFcs,af | cloud | IRFari | ERFcs,af | cloud |
| CH4 | -0.01 | 0.86 | 0.12 | -0.01 | 0.91 | -0.22 | 0.00 | 0.56 | -0.12 | -0.01 | 0.48 | -0.10 | 0.00 | 0.91 | -0.21 |
| HC | -0.02 | 0.02 | -0.18 | -0.02 | 0.22 | -0.14 | -0.01 | -0.02 | -0.08 | | | | -0.02 | 0.50 | -0.17 |
| N2O | -0.01 | 0.26 | 0.01 | | | | 0.00 | 0.41 | -0.09 | -0.01 | 0.24 | -0.00 | -0.00 | 0.23 | -0.03 |
| O3 | -0.02 | 0.16 | 0.07 | -0.04 | 0.49 | -0.18 | | | | | | | -0.00 | 0.24 | -0.18 |
| NOx | -0.03 | 0.10 | -0.05 | -0.02 | 0.25 | -0.09 | | | | | | | -0.01 | 0.03 | -0.04 |
| VOC | 0.00 | 0.13 | 0.20 | -0.02 | 0.18 | -0.08 | | | | | | | 0.004 | 0.17 | -0.2 |


**4.2.2 Ozone changes**
The ozone radiative forcing is diagnosed using a kernel to scale the 3D ozone changes based on Skeie et al. (2020).
This kernel includes stratospheric temperature adjustment, but not tropospheric adjustments so gives a SARF.
These are shown in Fig. 7. Corresponding changes in the tropospheric and stratospheric ozone columns are shown
in figure S5,  Increased $CH_4$ concentrations give a SARF for ozone produced by methane of 0.14±0.03 W m$^{-2}$,
anthropogenic NOx emissions and VOC (including CO) emissions give SARFs of 0.20±0.07 and 0.11±0.04 Wm$^{-}$
$^2$ respectively. The O3 experiment comprised both NOx and VOC emission changes. The SARF in this experiment
(0.31±0.05 Wm$^{-2}$) is close to the sum of the NOx and VOC experiments (0.30±0.05 Wm$^{-2}$ for the same set of
models) showing little non-linearity in the chemistry (Stevenson et al., 2013).
There is a larger variation across models in the stratospheric ozone depletion from halocarbons (-0.15±0.10 Wm$^{-2}$)
with UKESM1 having noticeably larger depletion as seen in Keeble et al. (2020) giving a SARF of -0.33 Wm$^{-2}$.
$N_2O$ causes some stratospheric ozone depletion in these models, mainly in the tropical upper stratosphere where
depletion causes a positive forcing (Skeie et al., 2020), and increases tropospheric ozone (Fig. S6) giving a small
net positive SARF (0.03±0.01  Wm$^{-2}$).

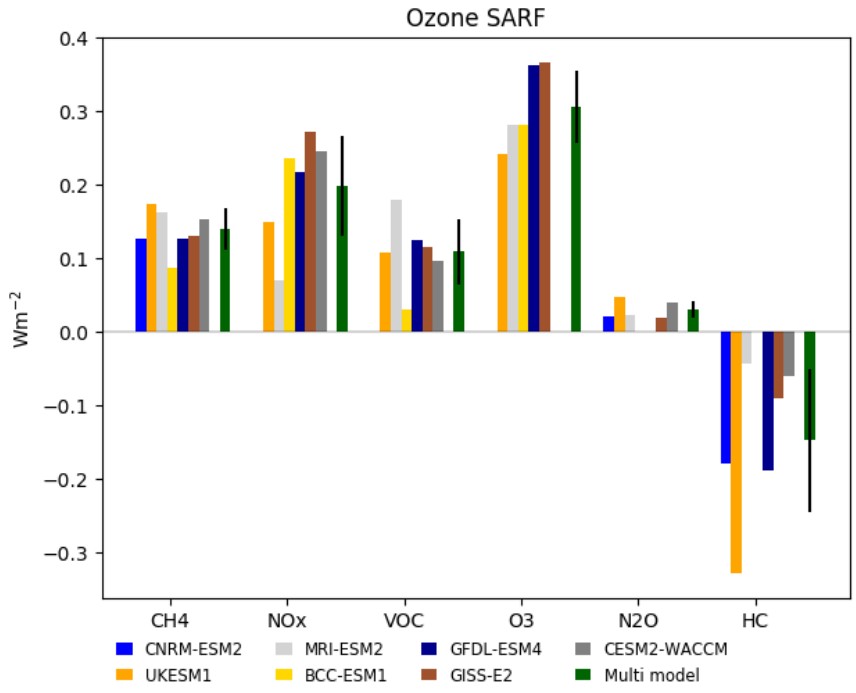


**Fig. 7 Changes in ozone stratospheric-temperature adjusted radiative forcing (SARF) for each experiment, diagnosed using kernels (see text). . Uncertainties for the multi model means are standard deviations across models.**

Methane oxidation also leads to water vapour production. Figure S6 shows increases in the stratosphere for the piClim-CH4 of up to 20% . The kernel analysis however finds very low radiative forcing associated with this increase ($-0.002\pm0.003$ Wm$^{-2}$).

**4.2.3 Comparison with greenhouse gas forcings**

The ERFs, ERFcs,af and SARFs diagnosed for the greenhouse gas changes (Fig. 6, Table 5) are compared with the expected greenhouse gas SARFs in Fig. 8. The expected SARFs from the well-mixed gases are given by Etminan et al. (2016) for CH$_4$ and N$_2$O, and by WMO (2018) for the halocarbons (the halocarbon changes are slightly different in each model). The expected SARFs from ozone changes are from Fig. 7.

For methane the ERFs are typically higher than the expected GHG SARF (except for CNRM-ESM2).The diagnosed ERFcs,af and SARF agree better with the expected SARF in UKESM1, BCC-ESM1 and CESM2-WACCM, but not in other models. For N$_2$O the modelled ERF is larger than the expected SARF for CNRM-ESM2-1 and CESM2-WACCM, this is explained by the rapid adjustments for CESM2-WACCM, but not for CNRM-ESM2. For halocarbons the stratospheric ozone depletion offsets the direct SARF and accounts for much of the spread in the model SARF, although the CNRM-ESM2-1 ERF and SARF is lower than expected. The modelled HC ERF for UKESM1 is strongly negative due to increased aerosol cloud interactions, (O'Connor et al., 2020a;Morgenstern et al., 2020) but removing cloud effects using the SARF or ERFcs,af agrees better with the expected value. The estimated ozone SARF from the NO$_X$, VOC and O3 experiments generally agrees with the model SARF and ERFcs,af. For CESM2-WACCM the ERF from the VOC experiment is zero, and the SARF negative even though the diagnosed ozone SARF is positive. For all experiments and models ERFcs,af is generally higher than the expected or diagnosed SARF (see section 3.3).

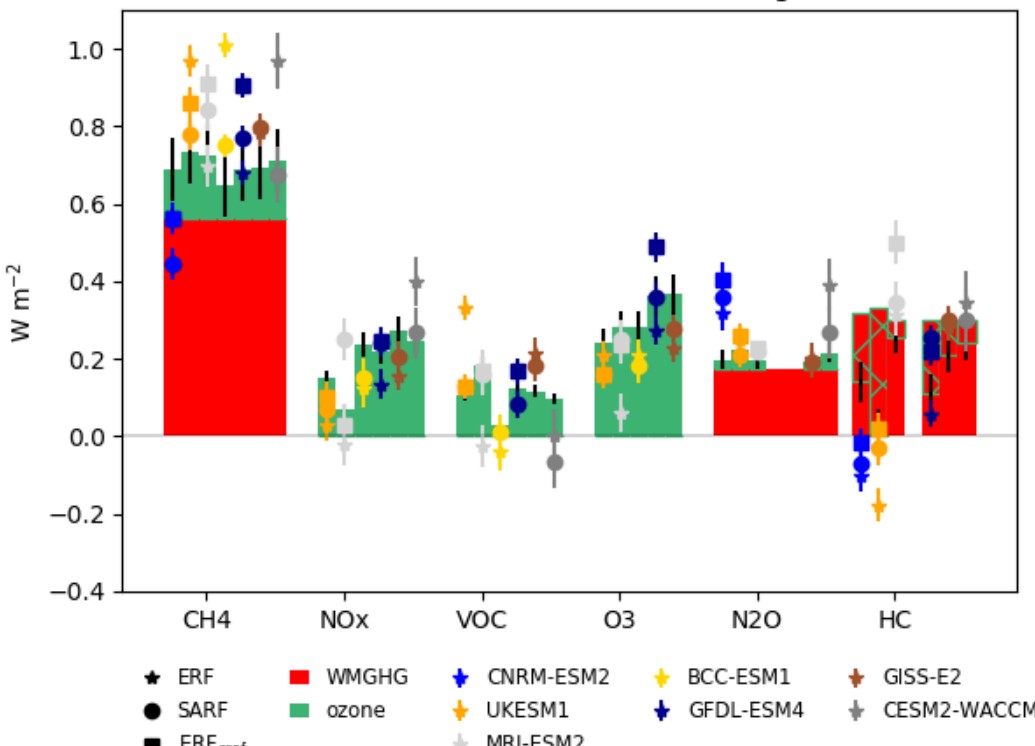

**1850 to 2014 radiative forcing**


**Fig. 8 Estimated SARF from the greenhouse gas changes (WMGHGs and ozone), using radiative efficiencies for the**
**WMGHGs and kernel calculations for ozone (see text). Hatched bars show decreases in ozone SARF. Symbols show**
**the modelled ERF, SARF and ERFcs,af (estimate of greenhouse gas clear-sky ERF). Uncertainties on the bars are due**
**to uncertainties in radiative efficiencies. Uncertainties on the symbols are errors in the mean due to interannual**
**variability in the model diagnostic.**
**4.2.4 Methane Lifetime**
In the CMIP6 setup the modelled methane concentrations do not respond to changes in oxidation rates. The
methane lifetime is diagnosed (which includes stratospheric loss to OH as parameterised within each model) and
assuming losses to chlorine oxidation and soil uptake of 11 and 30 Tg yr$^{-1}$ ((Saunois et al., 2020;Myhre et al.,
2013b) and this can be used to infer the methane changes that would be expected if methane were allowed to vary.
Fig. 9 shows the methane lifetime response is large and negative for $NO_x$ emissions, with a smaller positive change
for VOC emissions. Halocarbon concentration increases decrease the methane lifetime, as ozone depletions leads
to increased UV in the troposphere and increased methane loss to chlorine in the stratosphere (Stevenson et al.,
2020). N2O also decreases the methane lifetime by depleting ozone in the tropics although the effect is less than
for halocarbons. The O3 experiment has a significantly more negative effect (-27±9 %) than the sum of NOx and
VOC (-16±8 %) (uncertainties are multi-model standard deviation). This suggests significant non-additivity. Note
that a combined $CH_4$+NOx+VOC experiment is not available to test the additivity further.
The lifetime response to changing methane concentrations can be used to diagnose the methane lifetime feedback
factor *f* ((Fiore et al., 2009). The results here give *f*=1.32, 1.31, 1.43, 1.30, 1.26, 1.19 (mean 1.30±0.07) for
UKESM1, MRI-ESM2, BCC-ESM1, GFDL-ESM4, GISS-E2-1 and CESM-WACCM. This is in very good
agreement with AR5, although their values are starting from a year 2000 baseline rather than pre-industrial.


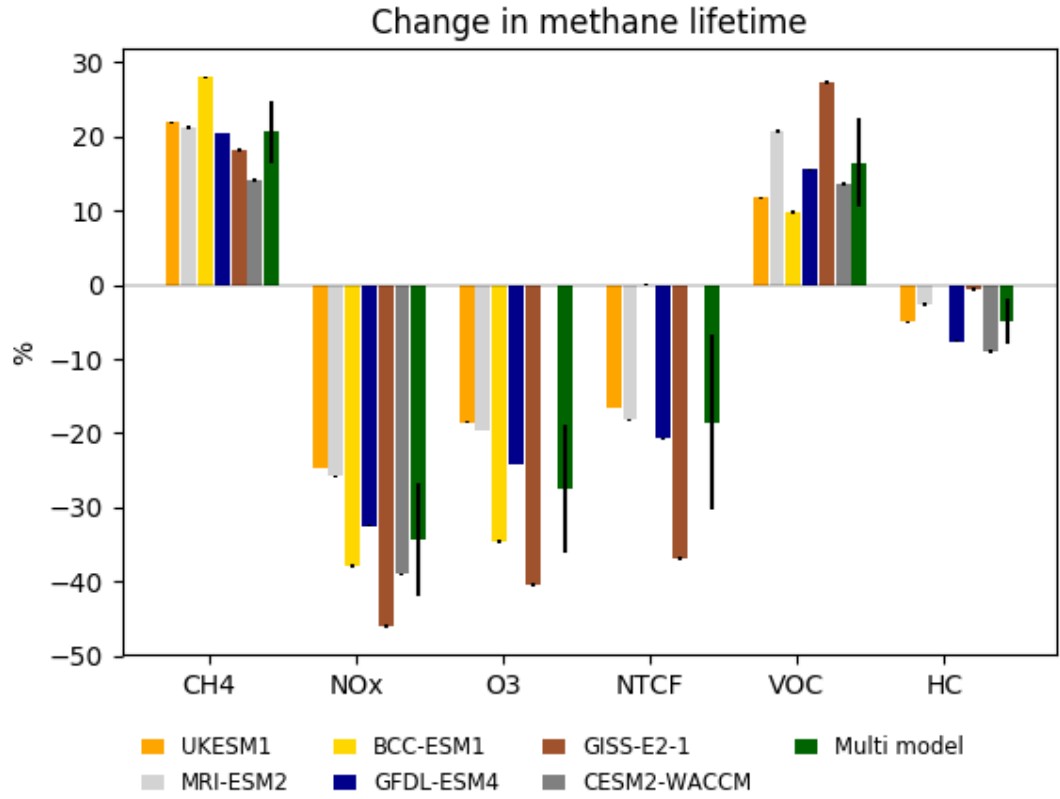


**Fig. 9 Changes in methane lifetime (%), for each experiment. Uncertainties for individual models are errors on the mean from interannual variability. Uncertainties for the multi model mean are standard deviations across models.**

### 4.2.5 Total ERFs

The methane lifetime changes can be converted to expected changes in concentration if methane were allowed to freely evolve following Fiore et al. (2009), using the *f*-factors appropriate to each model (section 3.3.4). The inferred radiative forcing is based on radiative efficiency of methane (Etminan et al., 2016). The methane changes also have implications for ozone production, so we assume an ozone SARF per ppb of $CH_4$ diagnosed for each model from section 4.2.

The breakdown of the information from the analyses above is shown in Fig. 10, using the SARF calculated for the gases (WMGHGs and ozone) and kernel-diagnosed cloud adjustments (which include aerosol cloud interactions). Direct contributions from the aerosols IRFari are shown for models where this is available. The contributions from methane lifetime changes have also been added to the diagnosed ERF as these aren't accounted for in the models. Differences between the diagnosed ERF (stars) and the sum of the components (crosses) then shows to what extent this decomposition into components can account for the modelled ERF. For many of the species, this breakdown is reasonable, and illustrates that cloud radiative effects can make significant contributions to the total radiative impacts of WMGHGs and ozone precursors. This analysis cannot distinguish between cloud effects due to changes in atmospheric temperature profiles or those due to increased cloud nucleation from aerosols.

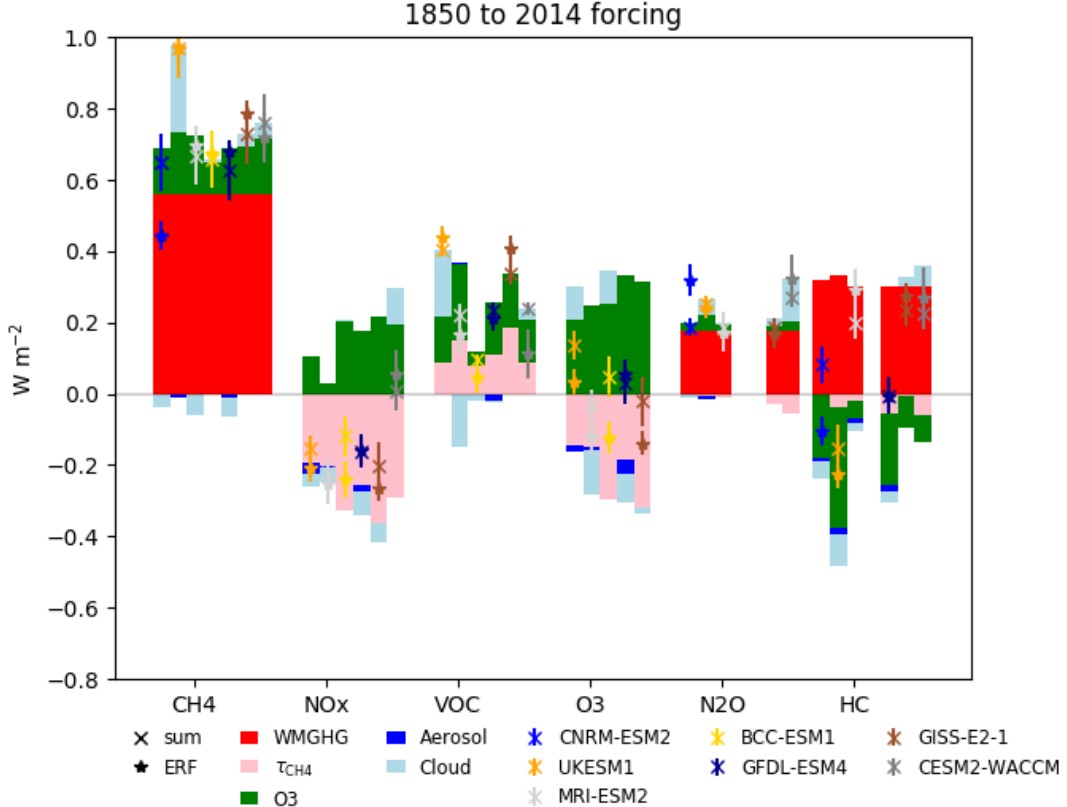

**Fig. 10 SARF for WMGHGs, ozone and diagnosed changes in methane. Model diagnosed direct aerosol RF and cloud**
**radiative effect. Crosses mark the sum of the five terms for each model. Stars mark the diagnosed ERF with the effect**
**of methane lifetime (on methane and ozone) added. Differences between stars and crosses shows undiagnosed**
**contributions. Uncertainties on the sum are mainly due to the uncertainties in the radiative efficiencies. Uncertainties**
**in the ERF are errors on the mean due to interannual variability. Note for CESM2-WACCM, BCC-ESM1, GISS-E2-1**
**the direct aerosol effect is unavailable.**
**5. Discussion**
For all of the species shown we see considerable variation in the calculated ERFs across the models, which is due
in part to differences in the model aerosol and chemistry schemes; not all models have interactive schemes for all
of the species, and whether or not chemistry is considered will impact the evolution of some of the aerosol species.
We can use the differences in model complexity from the multi-model approach together with the separation of
the effects of the various species in the individual AerChemMIP experiments to understand how the various
components contribute to the overall ERFs we have calculated.

**5.1 Aerosols**
The 1850-2014 multi-model mean and standard deviation of the ERFs for $SO_2$, OC and BC are: -1.03 +/- 0.37
$Wm^{-2}$ for $SO_2$, -0.25 +/- 0.09 $Wm^{-2}$ for OC, and 0.15 +/- 0.17 $Wm^{-2}$ for BC. The total ERF for the aerosols is -
1.01 +/- 0.25 $Wm^{-2}$, within the range of -1.65 to -0.6 $Wm^{-2}$ reported by (Bellouin et al., 2019).
The radiative kernels and double-call diagnostics are used to separate the direct and cloud effects of aerosols for
those models where all the relevant diagnostics are available. These two methods broadly agree on the cloud
contribution for the BC, $SO_2$ and OC experiments. We generally find a weaker total adjustment to black carbon
compared to other studies (Samset and Myhre, 2015;Stjern et al., 2017;Smith et al., 2018). The exceptions are
MIROC6 and GISS-E2-1. These previous studies used much larger changes in black carbon (up to 10 times)
which may cause non-linear effects such as self-lofting.
As the ISCCP cloud diagnostics become available for more of the CMIP6 models, it will be possible to do a direct
calculation of the cloud rapid adjustments using the kernels from (Zelinka et al., 2014) and compare those with
the adjustments calculated using the kernel difference method described in (Smith et al., 2018) and used here
(Section 3.2; see also figure 4 and figure S2 from Smith et al. (2020a)).
The radiative efficiencies per AOD calculated here are generally larger than those from the AeroCom Phase II
experiments (Myhre et al., 2013b), with the caveat that the models included here did not have fixed clouds, so
that indirect effects would be included.
The values diagnosed for the IRFari (for the models we have available diagnostics for) in CMIP6 are similar to
those from CMIP5 (Myhre et al., 2013a) where they reported values for sulfate of -0.4 (-0.6 to –0.2) $Wm^{-2}$
compared to our -0.36 (-0.19 to -0.49) $Wm^{-2}$ for the $SO_2$ experiment, for OC they found -0.09 (-0.16 to –0.03)
$Wm^{-2}$ compared to our value of -0.09 (-0.07 to -0.15) $Wm^{-2}$ and for BC they had +0.4 (+0.05 to +0.80) compared
to our value of 0.28 (0.13- 0.37) $Wm^{-2}$, so broadly the IRFari for the individual species agree with those found in
the previous set of models used in CMIP5.
The overall aerosol ERF from AR5 is reported as in the range -1.5 to 0.4 $Wm^{-2}$, compared to ERF values reported
here for the piClim-aer experiment in the range -0.7 to -1.47 $Wm^{-2}$.

## 629     5.2 Reactive greenhouse gases

The diagnosed ERFs from methane, $N_2O$, halocarbons and ozone precursors are: 0.75±0.10, 0.26±0.07, 0.12±0.21
and 0.20±0.07 W $m^{-2}$ (excluding CNRM-ESM2-1 for methane as it cannot represent the lower tropospheric ozone
changes, and excluding NorESM2 for all as it has no ozone chemistry). These compare with 0.79±0.13, 0.17±0.03,
0.18±0.15 and 0.22±0.14 W $m^{-2}$ for 1750-2011 from AR5 (Myhre et al., 2013a) - where the effects on methane
lifetime and $CO_2$ have been removed from the AR5 calculations, and the halocarbons are for CFCs and HCFCs
only. Section 4.2.5 shows that cloud effects can make a significant contribution to the overall ERF even for
WMGHGs. However, clouds cannot explain all the differences. The ERF for $N_2O$ is larger than estimated in AR5.
The ozone contribution here is estimated as 0.03±0.01 $Wm^{-2}$ whereas it was zero in AR5, but that does not explain
all the difference. The multi-model ERF for halocarbons is smaller than AR5, due to larger ozone depletion
although the models have a wide spread with some showing significantly lower ERFs and some significantly
higher due to varying strengths of ozone depletion in these models.
The estimated ozone SARFs from the changes in levels of methane, NOx and VOC from 1850 to 2014 are
0.14±0.03, 0.20±0.07, and 0.11±0.04 W $m^{-2}$ compared to 0.24±0.13, 0.14±0.09, and 0.11±0.05 W $m^{-2}$ in CMIP5
(Myhre et al., 2013a). The ozone from methane contribution is smaller, here only 25% of the direct Etminan et al.
(2016) methane SARF compared to 50% in AR5 (or 39% using the Etminan et al. (2016) formula). The NOx
contribution is larger in this study. The CMIP5 results were based on (Stevenson et al., 2013) in which species
were reduced from present day levels rather than being increased from pre-industrial levels. The NOx emission
changes are also larger for CMIP6 compared to CMIP5 (Hoesly et al. 2018). The sum of the ozone terms
($CH_4+N_2O+HC+O_3$) is $0.33\pm0.11$ $Wm^{-2}$, agreeing well with the total 1850-2014 ozone SARF of $0.35 \pm0.16$ $Wm^{-2}$
(1.s.d) from Skeie et al. (2020) which included a few additional models.

The overall effect of NTCF emissions (excluding methane and other WMGHGs) on the 1850-2014 ERF
experienced by models that include tropospheric chemistry is strongly negative ($-0.89\pm0.20$ $W\ m^{-2}$) due to the
dominance of the aerosol forcing over that from ozone. There is a large spread in the NTCF forcing due to the
different treatment of atmospheric chemistry within these models. Models without tropospheric and/or
stratospheric chemistry prescribe varying ozone levels which are not included in the NTCF experiment. Hence
the overall forcing experienced by these models due to ozone and aerosols will be different from that diagnosed
here.
**6. Conclusion**
The experimental setup and diagnostics in CMIP6 have allowed us for the first time to calculate the effective
radiative forcing (ERF) for present day reactive gas and aerosol concentrations and emissions in a range of Earth
system models. Quantifying the forcing in these models is an essential step to understanding their climate
responses.
This analysis also allows us to quantify the radiative responses to perturbations in individual species or groups of
species. These responses include physical adjustments to the imposed forcing as well as chemical adjustments and
adjustments related to the emissions of natural aerosols. The total adjustment is therefore a complex combination
of individual process, but the diagnosed ERF implicitly includes these and represents the overall forcing
experienced by the models.
We find that the ERF from well-mixed greenhouse gases (methane, nitrous oxide and halocarbons) has significant
contributions through their effects on ozone, aerosols and clouds, that vary strongly across Earth system models.
This indicates that Earth system processes need to be taken into account when understanding the contribution
WMGHGs have made to present climate and when projecting the climate effects of different WMGHG scenarios.
**7. Acknowledgements**
GT, WC, MM, FMO'C, DO, and MS acknowledge funding received from the European Union's Horizon 2020
research and innovation programme under grant agreement No 641816 (CRESCENDO).
D.O. and M.S. were also supported by the Research Council of Norway (grant no. 270061) and by the
Norwegian infrastructure for computational science (grant nos. NN9560K and NS9560K).
FMO'C and JPM were funded by the Met Office Hadley Centre Climate Programme funded by BEIS and Defra
(GA01101).
CS was supported by a NERC-IIASA Collaborative Research Fellowship (no. NE/T009381/1). GZ was
supported by the NZ government's Strategic Science Investment Fund (SSIF) through the NIWA programme
CACV. MD and NO were supported by the Japan Society for the Promotion of Science (grant numbers:
JP18H03363, JP18H05292, and JP20K04070), the Environment Research and Technology Development Fund
(JPMEERF20172003, JPMEERF20202003, and JPMEERF20205001) of the Environmental Restoration and
Conservation Agency of Japan, the Arctic Challenge for Sustainability II (ArCS II), Program Grant Number
JPMXD1420318865, and a grant for the Global Environmental Research Coordination System from the
Ministry of the Environment, Japan. T. T. was supported by the supercomputer system of the National Institute
for Environmental Studies, Japan, and JSPS KAKENHI Grant Number JP19H05669.
R.B.S. and G.M. were funded through the Norwegian Research Council project KEYCLIM (grant number
295046) and the European Union's Horizon 2020 Research and Innovation Programme under Grant Agreement
820829 (CONSTRAIN).
The CESM project is supported primarily by the National Science Foundation. This material is based upon work
supported by the National Center for Atmospheric Research, which is a major facility sponsored by the NSF
under Cooperative Agreement No. 1852977. Computing and data storage resources, including the Cheyenne
supercomputer (doi:10.5065/D6RX99HX), were provided by the Computational and Information Systems
Laboratory (CISL) at NCAR.
We acknowledge the World Climate Research Programme, which, through its Working Group on Coupled
Modelling, coordinated and promoted CMIP6. We thank the climate modeling groups for producing and making
available their model output, the Earth System Grid Federation (ESGF) for archiving the data and providing
access, and the multiple funding agencies who support CMIP6 and ESGF.

## 8. Author Contributions

Manuscript preparation was done by GDT, WJC, RJK, DO and additional contributions from all co-authors.
Model simulations were set up, reviewed and/or ran by RChG, DO, FMO'C, NLA, MD, LE, LH, J-FL, MMichou,
MMills, JM, PN, VN, NO, MS, TT, ST, TW, GZ, JZ. Analysis was carried out by GT, WC, RK, DO, RS.

## 9. Competing Interests

The authors declare that they have no conflict of interest.

## 10. Data Availability

All data from the various earth system models used in this paper are available on the Earth System Grid Federation
Website, and can be downloaded from there. https://esgf-index1.ceda.ac.uk/search/cmip6-ceda/

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
