# Peer review of "Effective radiative forcing from emissions of reactive gases and"

_Atmospheric Chemistry and Physics, 2019_

## Referee Comment (RC1) · Anonymous Referee #1 · 3 Apr 2020

This paper examines the effective radiative forcing (ERF) of reactive gases (e.g., CH4, NOx, VOC, O3, N2O, HC, NH3) and aerosols (e.g., BC, OC, SO2) to the climate system using multi-model output from the AerChemMIP experiments of the CMIP6 project. The contribution of each species to the total ERF is decomposed, and the differences of the calculated ERFs by various models are discussed. The paper is overall well written and easy to follow. I have some minor comments for the authors to consider before publication:

1. It is not clear how many ensemble members are used for each model. Can you please clarify this?

[Figure]

2. Fixed SST and sea ice are used in the ERF simulations. Is it the climatological SST of the 1850s?

3. The description of Eq. 5 is a bit vague. You may want to add "ERFaci" to line 240 after "The effect of the aerosol on cloud radiative forcing".

4. "A_trop" is used in text while "RA Trop" is used in Fig. 1. It's better to be consistent.

5. Line 373, "Fig. 6"→ "Fig. 5". And can you explain a bit more about how the total AOD is used to calculate the sum of the scaled ERFs?

6. Table 6, second row, "Nox"→ "NOx"

7. Line 465, please remove the brackets around "O'Connorl F. M. 2019"

8. Error bars are used in Fig. 9-10 to quantify the uncertainties due to interannual variability of model diagnostics. Is it possible to apply similar approach (error bars) to other figures where data are available?

9. Line 507, "RFs"→ "ERFs"

10. Line 525, "+/-"→ "±", please also add explanation about the numbers follow the sign, for instance, is it a standard deviation of multi-model output?

11. Lines 541-542, the overall aerosol ERFari from AR5 (-1.5 ∼ 0.4 Wm-2) is much larger than values reported here (-0.16 ∼ 0.03 Wm-2). Can you add some discussion about the differences?

12. Lines 546-550, redundant

---

## Referee Comment (RC2) · Anonymous Referee #2 · 11 May 2020

I've read the paper "Effective radiative forcing and adjustments in CMIP6 models" by Smith et al. The paper presents an important analysis of effective radiative forcing from CMIP6 RFMIP experiment. The analysis is sound, but I have a few suggestions for improvement.

The text could use some smoothing overall. In the initial sections in particular, the text is sometimes dis-jointed, with concepts and methods partially introduced, but then not really described until a later section.

Hopefully the results can be updated with model data made available since 12/19/19.

The paper could use a little more discussion of different ERF calculation methods.

[Figure]

This could usefully go in the conclusions section. How might the results presented here change if different methodologies and/or definitions were used. For example, the method and definition of effective forcing excludes "fast" responses related to sea-ice and SST changes. How potentially important are these for the different forcing components?

It would be helpful if the authors also discussed a bit more in the last section the potential limitations of this work and what future work might be warranted to improve forcing estimates.

It would be useful also to comment on how

Need to clarify if in numerical statements such as these: "ERF is $-1.04$ ($\pm0.23$) W m$-2$ from 12 models." the 1 standard deviation uncertainty provided is (I presume) just due to combining the individual model results, and that no additional uncertainty is included. (e.g., this certainly is an underestimate of the total uncertainty, although by how much it is not clear - as noted in next comment.)

My major quantitative comment is that no uncertainty bars are included for the individual model results. While I realize that inter-model differences are likely do dominate the overall uncertainty estimates, but this may not always be the case and some more analysis should be performed. In particular, it would be useful to better understand which model results are significantly different(statistically) from other models. I consider this a critical issue (thus "major revision"), however I don't believe that addressing this necessarily will entail a huge amount of additional work (assuming the authors current implicit assumption that uncertainty in the individual model results is small enough to be neglected is validated).

It is not sufficient to simply "Using 30 year timeslices generally results in standard absolute errors of less than 0.1 W m$-2$ (Forster et al., 2016)." How about in cases where the errors are larger than 0.1 W m$-2$? There are many models here, some are noisier than others, or have weaker signals.

There is some level of uncertainty in the data fits used in this method. This should should be easily quantifiable by the authors and should be reported (perhaps largely in SI). There is also potential biases and uncertainty introduced by using this specific methods - this should at least be discussed (and in some cases quantified, see specific comment on the kernel method below).

For example, if the combined methodological uncertainty of the individual model results was 0.1 W m-2 (as stated, but not shown), then, assuming normal distributions and independence, $\pm 0.23$ becomes $\pm 0.25$. So a fairly small change. However if the methodological uncertainty was larger, this would not necessarily be such a small change.

Some estimate of uncertainty is necessary to better justify the neglect of uncertainty in the individual model results. An example showing the uncertainty in fit for each individual model result, for at least a couple key examples (e.g., CO2 aerosol effects), would be useful to determine if there are some model results where the uncertainty is higher than average.

I would also suspect that for some effects, particularly related to cloud responses, the uncertainty might be larger than for other effects. If so, some of the differences between models might not be statistically signifiant, particularly where forcing is small. This would be important to know.

Also, numerical values for all main results, by model, e.g., in Figue 1 need to be provided in the SI. Results by forcing component and model should also be provided as data files (e.g. .csv file) as part of the supplemental material (GMD does make it obvious that this can be done, but many articles now provide such information). The results from this paper are likely to be widely used, and providing data files will help immensely to avoid transcription errors and save many person-hours in aggregate.

Only data sources (RFMIP and HadGEM3 kernels) are cited in the data availability. sector. While this does not seem to be a requirement of this journal (editors can clarify), it is preferable in this era that the codes used to perform the main calculations

Interactive
comment

were also made available along with the journal paper and archived in an appropriate archival repository. At a minimum, as mentioned above, the numerical results from the paper need to be made available in numerical form in either the paper supplement or an archival repository (note that an author's or departmental web site is no longer acceptable for this purpose.).

Specific comments:

pg 4, line 77 this text "The experiments and results presented in this study follow on from the assessment" is a bit confusing. This really isn't a follow on study, as such, but the two results are certainly related.

pg 5. The bulleted list here needs an introductory phrase to the effect of "In this work:"

Line 167 " With the exception of stratospheric temperature adjustments to greenhouse-gas forcing, structural differences introduced by using different kernels are well within 0.1 W m−2" Clarify - does this mean that the kernel method introduces this level of error, or that applying a kernel derived from one model to results from a different model introduces this magnitude of error? There are many models used here, I would think it is quite possible that some might have behavior that differs more widely from the kernel that is used? (Or is there some physical reason why this would not be the case?)

It would be useful to add a couple sentences to section 4.2.4 Kernel masking to walk the reader through what is happening in Equation 5.

line 257 "As shown in fig. 1 and discussed in section 5.1, ERF is approximately equal to RF for CO2, and we apply the Etminan formula to ERF". Reword since this is section 5.1.

line 324 "range of aerosol ERF estimates for 2014 versus" re-word - forcing is always relative to a specified reference point, so this doesn't quite make sense.

line 334 clarify if CNRM-ESM2-1 and UKESM1-0-LL were not in the PDRMIP models.

line 336 was MRI-ESM2-0 in PDRMIP?

line 345 edit to clarify, which experiment this is referring to. e.g., "The increase in cloud albedo leads to a strong negative SW radiative effect in the RFMIP all aerosol experiment"

line 369 something wrong with wording here "It may be the case that aerosol forcing over the whole historical transient aerosol forcing is "

end of Section 5.3.3 Please comment on how the lack of ice cloud interactions in most of the models impacts the multi-model mean forcing results. If, for example, the other models had a response to ice clouds similar to the "four models that include ice cloud interactions", would the forcing shift noticeably one way or the other?

Line 411 " ERFari, the APRP and double-call methods sometimes disagree". For those six models averaged together, is there a net difference between the two methods for ERFari (which might indicate a bias in one or the other methodologies), or are the differences of different sign between the models and cancel?

Line 447 - Is this referring to tropospheric and/or stratospheric ozone? Did all models include tropospheric ozone changes? If not, that would impact the implied ozone forcing values (e.g., actual ozone forcing would be larger). Clarify.
* * *

---

## Editor Comment (EC1) · Hailong Wang (Editor) · 17 May 2020

Rev of acp-2019-1205
Thornhill et al. "Effective Radiative forcing from emissions 1 of reactive gases and aerosols – a multimodel comparison"

This paper is a very important contribution to climate and atmospheric chemistry studies.  It is critical for the current IPCC AR6 assessment.  My apologies to the authors for the delay in my review – surprisingly, these isolation times do not make it easier to review. I am rushing to get this out and so there may be typos in this review.

AerChemMIP is a very important project that is trying to make sense of a complex, nonlinear system of chemistry and clouds.  The design has some serious flaws, and we all knew that as it was being developed since there were obvious limitations in the number and complexity of experiments.  First, there is the nonlinearity, which comes our clearly in these results:  Exp(A) plus Exp(B) does not equal Exp(A+B).  Further, and because of the nonlinear nature, the choice of reference atmosphere (1850) will produce quite different results than another (2014).  Personally I would have much preferred to use the 2014 atmosphere as the reference atmosphere (at least we can compare the models to measurements) and then subtract the emission or concentration changes from 1850 to 2014.   So, we live with AerChemMIP and use it.  The analysis here is very good, but needs to develop more of an "assessment" view when reporting final numbers.  They should reflect some (subjective) adjustment of values (for non-linearity) or bias (from 1850 atmosphere).  Simply reporting the model average for Exp(A) and then Exp(B) will lead anyone who uses this paper to assume that the combined effect is the sum.  That would not be good for either policy or attribution work.  If the sum is the best answer (I would think that so), then the contributions of components may need ot be expressed in % of total rather than in W/m2.  I do not recommend a substantial rewrite, but rather a self-assessment by the author team of how to use these results.

The abstract starts off being very careful and clear, but then I get lost between emissions and composition change.  For example, SO2 is clearly an emission-based ERF, while CH4 is an abundance-based ERF.  VOCs & NOx are obviously emissions (having direct RF) but O3 is not emitted.  The HC (L40: halocarbons? as opposed to NMHC?) are negative presumably from the ozone depletion, but this is an odd way to list the CFCs.  Methane is singled out at the end of the abstract as increasing ozone, but ozone is listed separately.  Is the methane based only on the change in CH4 concentration? Further, I suspect that the methane includes stratospheric water vapor which is indirect as is ozone, so why the separate listings.  This is just inconsistent and you really need to present the framework and rules for partitioning and assigning ERF.  Basically, AerChemMIP is a complicated set of overlapping experiments and thus the abstract with simple results is very difficult to write correctly.  Try to keep it clear and simple.

L53- aerosols are chemically reactive. try again – aerosols and gases that …

L56- climate feedbacks on natural emissions is a tough one for the non-CO2 species.  I do not think AerChemMIP did anything on that.

L68- "conditions" do you mean SST, I doubt you prescribed different chlorophyll or DMS?

L69- why aerosols here? It seems more like and ie than and eg, why not just say aerosols and gases.

L71- again the contrast of 'aerosols and chemistry' is really a poor description of AerChemMIP. Aerosols are a chemically reactive species (most of them are, they were created by gas phase chemistry). This Intro really must have a better inclusive discussion of greenhouse gases (including ozone), of indirect greenhouse gases (CO) and of aerosol (primary and secondary). This is at odds with the content of AerChemMIP.

L90- again, every time I read this, it sounds odd and misleading: " aerosols and reactive chemistry" are not opposites.

L93- again, this phrasing sounds wrong: " anthropogenic and reactive species" is this tow separate species or is it 'both'

L103- what does 'down to' mean? from 0.001 to 560 hPa?.

L107- typo 'is includes'

L102-L165 This information should really be in a table somewhere, not in the text. Focus on the results. In fact, most of this is already in Table 1.

L172-173- You should say 'emissions' with the NTCFs, also you should note that methane is an SLCF, which your statement seems to preclude. Also, I thought that SLCF was new preferred, but…

L174- There is a serious problem with the AerChemMIP as defined and we realized this at the time, but did not address it: viz. because of the large changes in atmospheric chemistry and oxidants between 1850 and 2014 (including the ozone depletion), it is not clear that the effect of today's NTCF emissions in today's atmosphere are anywhere close to those calculated here for the PI atmosphere. This needs to be addressed when compiling results and clearly adds to the uncertainty. The reason why it is dangerous is that it could totally misrepresent the magnitude of the response if we were to cut NTCF today.

L200- "ocean state" implies much more than SSTs – do you really mean that.

L235- I am confused here, it seems like the direct forcing would include, not ignore, the absorption and scatting by aerosols. Is this a typo.

L239- this section is very confusing as written. I am sure it is simpler that it seems but some of the writing seems incorrect. In this line surely you mean the aerosol direct radiative effect, since "radiative effects due only to aerosols" would imply both ari and aci. The notation for ERFcs,af is inconsistent between eqn and text (comma or not). Eqns 4 & 5 sum to give not ERF, so where is this missing 'surface albedo' term (ERF-ERFcsaf) and does it matter? It must certainly be counted as an aerosol ERF. [OK, I see this in Table 3 later, is the cs,af just noise?]

L254ff- This is odd, you said just above the SARF is calculated from ERF – A_trop, so of course this should give 0% difference. Are you just checking the math? Also, with SARF calculated as a residual term, and the ability to denote and sum all the A_trop being highly uncertain, SARF would not appear to be very certain. In fact the SARF term would depend on the models' ability to diagnose A_trop correctly.

L260- With BC, the long-standing problem is that some models get far too much in the stratosphere and that would cause a very large SARF.

L285- in this figure and some others, please define carefully what the shorthand for the terms means.

L296ff- This is a very good discussion of the aerosol components!

L321- This would be a much better lead off to the aerosol section and analysis, begin with the big picture before the weeds.

L343 – just put this table into the figure, it is just a summary of the bars anyway.

L354- Please, stop wasting space, Table 5 and Fig 4 has the same information. If you want to show AOD, then add it to the figure.

L370ff & Fig 5- I do not understand the purpose of this AOD scaling, it really does little to help. The figure shows the key data: ERF from parts vs ERF from all. The ERF-parts consistently over accounts for the combined ERF. This is as expected wince the cloud effects are largest in a pi atmosphere with little background aerosols. So herein lies one of the fundamental problems with the AerChemMIP that must be acknowledged and accounted for. I am not sure that scaling by AOD is any more justified than just scaling the indirect to match – An interesting question is whether the ERF-ari sum of the parts equals the whole? Keep this fudge factor simple since there is no correct way to do this.

L393- Again, this title jars a bit, SO2 and NH3 are reactive gases.

L406- How big is the error in GHGas ERF if one ignores clouds? I would think large.

Fig 6 & Table 6, can easily combine and understand the Std Dev better.

L419- Here is a case where some assessment is due as to how well these model simulations are accurate in the sense of including all the effects. As noted the N2O-O3 link is important and missing in some, and the other key link between N2O and tropospheric OH and methane (and maybe aerosols) is established but missing here.

L431-441- This is a very interesting and important discussion about the additivity of the components. I suspect that the CH4 result is similarly affected.
BTW, where is the effect of stratospheric H2o form CH4 noted or counted?

L480- Yes halocarbons, but also N2O, and N2O may be more important since it depletes ozone in the tropics (there are papers on this). I do not wee what difference stratospheric Cl will make on the total lifetime – hopefully the Stevenson paper becomes referencable. Also, you need to be careful here since the methane feedback factors ff, apply to the PI atmosphere, and that is different from the present, particularly lower CO and NOx…. . The feedback factor used for GWPs etc, is the current one, not the PI one, so these results should NOT be used to change any previous assessments.

Fig 11 is really hard to understand or see clearly, it will need a cleanup.

L525- Here is a very important statement and I am not sure that you have put together all the reasonable uncertainties or non-linear scaling. The individual components here must be adjusted to recognize that the total ERF (with all simultaneously) is much less than the sum of the components. Thus you cannot recommend the individual results without scaling and without increasing the uncertainty.

L561- Same as above. The individual values will not sum correctly and so these do NOT reflect the ERF of NOx emissions as we progress from 1850 to 2014. Thus they should not be used as part of an assessment until they are more critically evaluated and put in context.
These are all very important AerChemMIP results, and the analysis here is highly valuable, but their use in attribution and related studies should reflect the bias and uncertainties in combining nonlinear parts that were calculated separately, and in basing these all on a pi-atmosphere.

---

## Author Comment (AC1) · 3 Jul 2020

Responses to reviews of 'Effective Radiative forcing from emissions of reactive gases and aerosols – a multimodel comparison' by G. Thornhill et al

We would like to thank the two referees for their helpful comments and suggestions. Our responses to the reviewers comments are below – reviewer comments in black, our responses in red.

**Anonymous Referee #1**

This paper examines the effective radiative forcing (ERF) of reactive gases (e.g., CH4, NOx, VOC, O3, N2O, HC, NH3) and aerosols (e.g., BC, OC, SO2) to the climate system using multi-model output from the AerChemMIP experiments of the CMIP6 project. The contribution of each species to the total ERF is decomposed, and the differences of the calculated ERFs by various models are discussed. The paper is overall well written and easy to follow. I have some minor comments for the authors to consider before publication:

1. It is not clear how many ensemble members are used for each model. Can you please clarify this?
In most cases there was a single member – this will be clarified.

2. Fixed SST and sea ice are used in the ERF simulations. Is it the climatological SST of the 1850s?
These were the fixed SSTs from the Pre-Industrial run – mentioned in L. 174

3. The description of Eq. 5 is a bit vague. You may want to add "ERFaci" to line 240 after "The effect of the aerosol on cloud radiative forcing".
This will be clarified with an improved description.

4. "A_trop" is used in text while "RA Trop" is used in Fig. 1. It's better to be consistent.
This be corrected to keep the reference to the quantity consistent

5. Line 373, "Fig. 6"! "Fig. 5". And can you explain a bit more about how the total AOD is used to calculate the sum of the scaled ERFs?
We will correct the Fig. number reference and add more detail on how the scaling is calculated.

6.Table 6, second row, "Nox"! "NOx"
We will correct the typo

7. Line 465, please remove the brackets around "O'Connorl F. M. 2019"
Reference will be fixed.

8. Error bars are used in Fig. 9-10 to quantify the uncertainties due to interannual variability of model diagnostics. Is it possible to apply similar approach (error bars) to other figures where data are available?
We will add error bars where appropriate and we have the data.

9. Line 507, "RFs"! "ERFs"
This will be corrected to ERFs

10. Line 525, "+/-"! "_", please also add explanation about the numbers follow the

sign, for instance, is it a standard deviation of multi-model output?

This will be clarified.

11. Lines 541-542, the overall aerosol ERFari from AR5 (-1.5 _ 0.4 Wm-2) is much larger than values reported here (-0.16 _ 0.03 Wm-2). Can you add some discussion about the differences?

Yes, discussion will be added here on the differences between the AR5 ERFari and that reported here (a difference in definitions, I think)

12. Lines 546-550, redundant

Redundant lines will be removed.
* * *
**Anonymous Referee #2**

Thornhill et al. "Effective Radiative forcing from emissions 1 of reactive gases and aerosols – a multimodel comparison"
This paper is a very important contribution to climate and atmospheric chemistry studies. It is critical for the current IPCC AR6 assessment. My apologies to the authors for the delay in my review – surprisingly, these isolation times do not make it easier to review. I am rushing to get this out and so there may be typos in this review.
AerChemMIP is a very important project that is trying to make sense of a complex, nonlinear system of chemistry and clouds. The design has some serious flaws, and we all knew that as it was being developed since there were obvious limitations in the number and complexity of experiments. First, there is the nonlinearity, which comes out clearly in these results: Exp(A) plus Exp(B) does not equal Exp(A+B). Further, and because of the nonlinear nature, the choice of reference atmosphere (1850) will produce quite different results than another (2014). Personally I would have much preferred to use the 2014 atmosphere as the reference atmosphere (at least we can compare the models to measurements) and then subtract the emission or concentration changes from 1850 to 2014. So, we live with AerChemMIP and use it. The analysis here is very good, but needs to develop more of an "assessment" view when reporting final numbers. They should reflect some (subjective) adjustment of values (for non-linearity) or bias (from 1850 atmosphere). Simply reporting the model average for Exp(A) and then Exp(B) will lead anyone who uses this paper to assume that the combined effect is the sum. That would not be good for either policy or attribution work. If the sum is the best answer (I would think that so), then the contributions of components may need to be expressed in % of total rather than in W/m2. I do not recommend a substantial rewrite, but rather a self-assessment by the author team of how to use these results.

We will make some assessment of the limitations and constraints of the experimental design, and also clarify the nonlinearity in the discussion of the results; we could add contributions in percent of total (where appropriate), if it improves the clarity.

The abstract starts off being very careful and clear, but then I get lost between emissions and composition change. For example, SO2 is clearly an emission-based ERF, while CH4 is an abundance-based ERF. VOCs & NOx are obviously emissions (having direct RF) but O3 is not emitted. The HC (L40: halocarbons? as opposed to NMHC?) are negative presumably from the ozone depletion, but this is an odd way to list the CFCs. Methane is singled out at the end of the abstract as increasing ozone, but ozone is listed separately. Is the methane based only on the change in CH4 concentration? Further, I suspect that the methane includes stratospheric water vapor which is indirect as is ozone, so why the separate listings. This is just inconsistent and you

really need to present the framework and rules for partitioning and assigning ERF. Basically, AerChemMIP is a complicated set of overlapping experiments and thus the abstract with simple results is very difficult to write correctly. Try to keep it clear and simple.

We will clarify the abstract and add comments on the frameworks used, and where results are emissions vs. concentration based.

L53- aerosols are chemically reactive. try again – aerosols and gases that …

This will be re-worded as "chemically-reactive gases".

L56- climate feedbacks on natural emissions is a tough one for the non-CO2 species. I do not think AerChemMIP did anything on that.

The discussion of natural emissions will be removed. These are covered in Thornhill et al. 2020.

L68- "conditions" do you mean SST, I doubt you prescribed different chlorophyll or DMS?

We will clarify which conditions are referred to here.

L69- why aerosols here? It seems more like and ie than and eg, why not just say aerosols and gases.

It is simply meant as an exemplar of what is perturbed – hence the eg., but could be re-worded to include 'aerosols or gases'

L71- again the contrast of 'aerosols and chemistry' is really a poor description of AerChemMIP. Aerosols are a chemically reactive species (most of them are, they were created by gas phase chemistry). This Intro really must have a better inclusive discussion of greenhouse gases (including ozone), of indirect greenhouse gases (CO) and of aerosol (primary and secondary). This is at odds with the content of AerChemMIP.

This section will be expanded to clarify the indirect roles of precursor species as recommended by the reviewer.

L90- again, every time I read this, it sounds odd and misleading: " aerosols and reactive chemistry" are not opposites.

This will be reworded as "aerosols and reactive gases"

L93- again, this phrasing sounds wrong: " anthropogenic and reactive species" is this two separate species or is it 'both'

This will be reworded as "aerosols and reactive gases"

L103- what does 'down to' mean? from 0.001 to 560 hPa?.

This will be clarified in terms of levels of the atmosphere..

L107- typo 'is includes'

This will be corrected.

L102-L165 This information should really be in a table somewhere, not in the text. Focus on the results. In fact, most of this is already in Table 1.

Agreed – although some text discussing how model differences may affect results is appropriate here, with references to Table 1 and Table S1 for detailed information.

L172-173- You should say 'emissions' with the NTCFs, also you should note that methane is an SLCF, which your statement seems to preclude. Also, I thought that SLCF was new preferred, but…

The use of NTCF was retained because the experiment was called piClim-NTCF, so this nomenclature was kept to aid in understanding which experiment was referred to. The comment in brackets L. 173 makes the point that this nomenclature has changed. Other references to the term NTCF can have SLCF added parenthetically to point out the change in nomenclature. The wording will be clarified to describe methane correctly.

L174- There is a serious problem with the AerChemMIP as defined and we realized this at the time, but did not address it: viz. because of the large changes in atmospheric chemistry and oxidants between 1850 and 2014 (including the ozone depletion), it is not clear that the effect of today's NTCF emissions in today's atmosphere are anywhere close to those calculated here for the PI atmosphere. This needs to be addressed when compiling results and clearly adds to the uncertainty. The reason why it is dangerous is that it could totally misrepresent the magnitude of the response if we were to cut NTCF today.
The reviewer has a good point here. We will add a discussion of the issue likely differences between perturbing emissions in a pre-industrial and present day atmosphere.

L200- "ocean state" implies much more than SSTs – do you really mean that.
This will be changed to "SST".
.
L235- I am confused here, it seems like the direct forcing would include, not ignore, the absorption and scatting by aerosols. Is this a typo.
Agreed, this is unclear as written - we will re-write this to make it clear that this is the difference between the aerosol-free flux and the total flux.

L239- this section is very confusing as written. I am sure it is simpler that it seems but some of the writing seems incorrect. In this line surely you mean the aerosol direct radiative effect, since "radiative effects due only to aerosols" would imply both ari and aci. The notation for ERFcs,af is inconsistent between eqn and text (comma or not). Eqns 4 & 5 sum to give not ERF, so where is this missing 'surface albedo' term (ERF-ERFcsaf) and does it matter? It must certainly be counted as an aerosol ERF. [OK, I see this in Table 3 later, is the cs,af just noise?]
Agreed – this section needs clarification in the writing and description, consistency in the subscripts will be corrected, and the surface albedo term eqn. will be added in to complete the definitions.

L254ff- This is odd, you said just above the SARF is calculated from ERF – A_trop, so of course this should give 0% difference. Are you just checking the math? Also, with SARF calculated as a residual term, and the ability to denote and sum all the A_trop being highly uncertain, SARF would not appear to be very certain. In fact the SARF term would depend on the models' ability to diagnose A_trop correctly.
I think we defined the SARF here as the ($IRF+A_{temp\_strat}$). There is the ERF as calculated from the kernels using the sum of the adjustments and the IRF, which is compared to the direct calculation of the ERF in 3.1.1. I will clarify what is being compared here.

L260- With BC, the long-standing problem is that some models get far too much in the stratosphere and that would cause a very large SARF.
We will check the vertical profiles of BC in these models.

L285- in this figure and some others, please define carefully what the shorthand for the terms means.
We will make sure that terms in the figures are properly defined.

L296ff- This is a very good discussion of the aerosol components!

Thank you, nice to hear what is good about the paper!

L321- This would be a much better lead off to the aerosol section and analysis, begin with the big picture before the weeds.
Agreed – the overall values should come before the effort at analysing and breaking down the components (or 'the weeds'…)

L343 – just put this table into the figure, it is just a summary of the bars anyway.
I think it is useful to have the multi-model mean numbers and errors separated out so the figure is not cluttered, but the table could go in the Supplemental materials, and the multimodel mean bar added to the plot.

L354- Please, stop wasting space, Table 5 and Fig 4 has the same information. If you want to show AOD, then add it to the figure.
We think the numbers are useful to have separately, but the table could be moved to the Supplemental material.

L370ff & Fig 5- I do not understand the purpose of this AOD scaling, it really does little to help. The figure shows the key data: ERF from parts vs ERF from all. The ERF-parts consistently over accounts for the combined ERF. This is as expected wince the cloud effects are largest in a pi atmosphere with little background aerosols. So herein lies one of the fundamental problems with the AerChemMIP that must be acknowledged and accounted for. I am not sure that scaling by AOD is any more justified than just scaling the indirect to match – An interesting question is whether the ERF-ari sum of the parts equals the whole? Keep this fudge factor simple since there is no correct way to do this.
The idea was to assess the breakdown of the overall ERF by the constituent aerosol, to get a sense of the relative importance of the different aerosol species in the combined piClim-aer experiment, and to illuminate model differences (if any) in this breakdown. It is imperfect, but we thought a useful additional data point.

L393- Again, this title jars a bit, SO2 and NH3 are reactive gases.
This will be renamed to reactive greenhouse gases

L406- How big is the error in GHGas ERF if one ignores clouds? I would think large.
We will add a discussion of the expected size of the cloud masking effect

Fig 6 & Table 6, can easily combine and understand the Std Dev better.
As noted above (L 354), having the multimodel mean numbers separated into a table makes it easier for the reader to use them– the table can be moved to the supplemental material and a multimodel mean bar added to the plot, together with std error bars.

L419- Here is a case where some assessment is due as to how well these model simulations are accurate in the sense of including all the effects. As noted the N2O-O3 link is important and missing in some, and the other key link between N2O and tropospheric OH and methane (and maybe aerosols) is established but missing here.
We will make it clearer as to which processes are included in which models. For the relevant impacts on stratospheric/tropospheric ozone, and methane lifetime, only the models with appropriate chemical processes are considered.

L431-441- This is a very interesting and important discussion about the additivity of the components. I suspect that the CH4 result is similarly affected.
BTW, where is the effect of stratospheric H2o form CH4 noted or counted?

The effect of stratospheric H2O is included with the WMGHG. It is not diagnosed separately.

L480- Yes halocarbons, but also N2O, and N2O may be more important since it depletes ozone in the tropics (there are papers on this). I do not see what difference stratospheric Cl will make on the total lifetime – hopefully the Stevenson paper becomes referencable. Also, you need to be careful here since the methane feedback factors ff, apply to the PI atmosphere, and that is different from the present, particularly lower CO and NOx…. . The feedback factor used for GWPs etc, is the current one, not the PI one, so these results should NOT be used to change any previous assessments.

We will mention the change due to N2O as well. We will clarify that the f factors are starting from a pre-industrial atmosphere.

Fig 11 is really hard to understand or see clearly, it will need a cleanup.
This figure will be revised to only show the multi-model mean.

L525- Here is a very important statement and I am not sure that you have put together all the reasonable uncertainties or non-linear scaling. The individual components here must be adjusted to recognize that the total ERF (with all simultaneously) is much less than the sum of the components. Thus you cannot recommend the individual results without scaling and without increasing the uncertainty.
This discussion will be revised to explain the differences between the individual components and the total, and the implications for uncertainty.

L561- Same as above. The individual values will not sum correctly and so these do NOT reflect the ERF of NOx emissions as we progress from 1850 to 2014. Thus they should not be used as part of an assessment until they are more critically evaluated and put in context.
These are all very important AerChemMIP results, and the analysis here is highly valuable, but their use in attribution and related studies should reflect the bias and uncertainties in combining nonlinear parts that were calculated separately, and in basing these all on a pi-atmosphere.

This discussion will be revised to explain the differences between the sum of NOx and VOC and the total. The ERFs are typically defined starting from a pre-industrial atmosphere, there is no unique way to reflect the ERF of NOx emissions as we progress from 1850 to 2014 (subtracting components from a present atmosphere would overestimate the ERF).

---

## Author Response (AR1)

Responses to reviews of 'Effective radiative forcing from emissions of reactive gases and
aerosols – a multi-model comparison' by G. Thornhill et al
We would like to thank the two referees for their helpful comments and suggestions.
Our responses to the reviewers comments are below – reviewer comments in black, our
responses in blue.
Changes and edits to the text are noted in blue.
**Anonymous Referee #1**
This paper examines the effective radiative forcing (ERF) of reactive gases (e.g., CH4,
NOx, VOC, O3, N2O, HC, NH3) and aerosols (e.g., BC, OC, SO2) to the climate system
using multi-model output from the AerChemMIP experiments of the CMIP6 project. The
contribution of each species to the total ERF is decomposed, and the differences of
the calculated ERFs by various models are discussed. The paper is overall well written
and easy to follow. I have some minor comments for the authors to consider before
publication:
1. It is not clear how many ensemble members are used for each model. Can you
please clarify this?
We have clarified that only one ensemble member was used in these experiments in Section 2.2
2. Fixed SST and sea ice are used in the ERF simulations. Is it the climatological SST
of the 1850s?
These were the fixed SSTs from the Pre-Industrial run – mentioned in L. 185.
3. The description of Eq. 5 is a bit vague. You may want to add "ERFaci" to line 240
after "The effect of the aerosol on cloud radiative forcing".
We have re-written this section and changed the nomenclature to be consistent with other work in
this field, so that the descriptions of each term is clear.
4. "A_trop" is used in text while "RA Trop" is used in Fig. 1. It's better to be consistent.
We have changed the nomenclature and definitions in the text so that we use $A_{total}$ in the text and
the Fig.1 for this quantity.
5. Line 373, "Fig. 6"! "Fig. 5". And can you explain a bit more about how the total
AOD is used to calculate the sum of the scaled ERFs?
We have corrected the figure number and have added additional explanation on how the AOD is
used in this scaling. There is also more detailed discussion in the Supplementary material.
6.Table 6, second row, "Nox"! "NOx"
The typo has been corrected.
7. Line 465, please remove the brackets around "O'Connorl F. M. 2019"
The reference has been fixed.
8. Error bars are used in Fig. 9-10 to quantify the uncertainties due to interannual
variability of model diagnostics. Is it possible to apply similar approach (error bars) to
other figures where data are available?

We have added error bars to Fig. 1 and Fig. 6 to show the S.E. for the ERF individual models (for the inter-annual variability) and the S.D. for the multi-model means.

9. Line 507, "RFs"! "ERFs"
This has been corrected.

10. Line 525, "+/-"! "_", please also add explanation about the numbers follow the sign, for instance, is it a standard deviation of multi-model output?
We have clarified where the numbers are standard error or standard deviation throughout the manuscript.

11. Lines 541-542, the overall aerosol ERFari from AR5 (-1.5 _ 0.4 Wm-2) is much larger than values reported here (-0.16 _ 0.03 Wm-2). Can you add some discussion about the differences?
In fact the comparison here was incorrect – the AR5 number quoted is the total ERF, not the IRFari (in the original manuscript this was referred to as the ERFari), so this has been corrected.

12. Lines 546-550, redundant
The redundant lines have been removed.
* * *
**Anonymous Referee #2**

 Thornhill et al. "Effective Radiative forcing from emissions 1 of reactive gases and aerosols – a multimodel comparison"
This paper is a very important contribution to climate and atmospheric chemistry studies. It is critical for the current IPCC AR6 assessment. My apologies to the authors for the delay in my review – surprisingly, these isolation times do not make it easier to review. I am rushing to get this out and so there may be typos in this review.
AerChemMIP is a very important project that is trying to make sense of a complex, nonlinear system of chemistry and clouds. The design has some serious flaws, and we all knew that as it was being developed since there were obvious limitations in the number and complexity of experiments. First, there is the nonlinearity, which comes out clearly in these results: Exp(A) plus Exp(B) does not equal Exp(A+B). Further, and because of the nonlinear nature, the choice of reference atmosphere (1850) will produce quite different results than another (2014). Personally I would have much preferred to use the 2014 atmosphere as the reference atmosphere (at least we can compare the models to measurements) and then subtract the emission or concentration changes from 1850 to 2014. So, we live with AerChemMIP and use it. The analysis here is very good, but needs to develop more of an "assessment" view when reporting final numbers. They should reflect some (subjective) adjustment of values (for non-linearity) or bias (from 1850 atmosphere). Simply reporting the model average for Exp(A) and then Exp(B) will lead anyone who uses this paper to assume that the combined effect is the sum. That would not be good for either policy or attribution work. If the sum is the best answer (I would think that so), then the contributions of components may need to be expressed in % of total rather than in W/m2. I do not recommend a substantial rewrite, but rather a self-assessment by the author team of how to use these results.
We have added discussion of the non-linearity and where that would affect a simplistic interpretation of combined results in the relevant sections, and we have added comments on the limitations of the experimental design.

The abstract starts off being very careful and clear, but then I get lost between emissions and
composition change. For example, SO2 is clearly an emission-based ERF, while CH4 is an
abundance-based ERF. VOCs & NOx are obviously emissions (having direct RF) but O3 is not
emitted. The HC (L40: halocarbons? as opposed to NMHC?) are negative presumably from the
ozone depletion, but this is an odd way to list the CFCs. Methane is singled out at the end of the
abstract as increasing ozone, but ozone is listed separately. Is the methane based only on the
change in CH4 concentration? Further, I suspect that the methane includes stratospheric water
vapor which is indirect as is ozone, so why the separate listings. This is just inconsistent and you
really need to present the framework and rules for partitioning and assigning ERF. Basically,
AerChemMIP is a complicated set of overlapping experiments and thus the abstract with simple
results is very difficult to write correctly. Try to keep it clear and simple.
We have added text to the abstract to include a better description of the relationship between the
ozone pre-cursors and the ERFs calculated.

L53- aerosols are chemically reactive. try again – aerosols and gases that …
We have re-worded this as chemically reactive gases to make the distinction clearer.

L56- climate feedbacks on natural emissions is a tough one for the non-CO2 species. I do not
think AerChemMIP did anything on that.
The discussion of natural emissions has been removed. These are covered in Thornhill et al.
2020.

L68- "conditions" do you mean SST, I doubt you prescribed different chlorophyll or DMS?
We have clarified which conditions are meant here (SSTs and sea-ice).

L69- why aerosols here? It seems more like and ie than and eg, why not just say aerosols and
gases.
This has been reworded.

L71- again the contrast of 'aerosols and chemistry' is really a poor description of AerChemMIP.
Aerosols are a chemically reactive species (most of them are, they were created by gas phase
chemistry). This Intro really must have a better inclusive discussion of greenhouse gases
(including ozone), of indirect greenhouse gases (CO) and of aerosol (primary and secondary).
This is at odds with the content of AerChemMIP.
This section has been expanded to clarify the indirect roles of precursor species as recommended
by the reviewer.

L90- again, every time I read this, it sounds odd and misleading: " aerosols and reactive
chemistry" are not opposites.
This has been reworded as "aerosols and reactive gases".

L93- again, this phrasing sounds wrong: " anthropogenic and reactive species" is this two
separate species or is it 'both'
This has been reworded as "aerosols and reactive gases"

L103- what does 'down to' mean? from 0.001 to 560 hPa?.
This has been reworded to clarify the levels of the atmosphere.

L107- typo 'is includes'

*Typo has been corrected.*

L102-L165 This information should really be in a table somewhere, not in the text. Focus on the
results. In fact, most of this is already in Table 1.
*The summary of different model aerosol and chemistry modules has been retained in the text, and*
*the details moved to Supplementary Table S1.*

L172-173- You should say 'emissions' with the NTCFs, also you should note that methane is an
SLCF, which your statement seems to preclude. Also, I thought that SLCF was new preferred,
but…
*The use of NTCF was retained because the experiment was called piClim-NTCF, so this*
*nomenclature was kept to aid in understanding which experiment was referred to. We state that*
*the point that this nomenclature has changed and that SLCF is now preferred. In the piClim-*
*NTCF experiment in AerChemMIP methane is deliberately excluded as part of the experimental*
*design.*

L174- There is a serious problem with the AerChemMIP as defined and we realized this at the
time, but did not address it: viz. because of the large changes in atmospheric chemistry and
oxidants between 1850 and 2014 (including the ozone depletion), it is not clear that the effect of
today's NTCF emissions in today's atmosphere are anywhere close to those calculated here for the
PI atmosphere. This needs to be addressed when compiling results and clearly adds to the
uncertainty. The reason why it is dangerous is that it could totally misrepresent the magnitude of
the response if we were to cut NTCF today.
*The reviewer has a good point here. We have added discussion of the issue likely differences*
*between perturbing emissions in a pre-industrial and present day atmosphere, and commented in*
*places where this would influence the results.*

L200- "ocean state" implies much more than SSTs – do you really mean that.
*This has been corrected.*

.

L235- I am confused here, it seems like the direct forcing would include, not ignore, the
absorption and scatting by aerosols. Is this a typo.
*This was a typo, and the section has been re-written in order to explain the method more clearly.*

L239- this section is very confusing as written. I am sure it is simpler that it seems but some of
the writing seems incorrect. In this line surely you mean the aerosol direct radiative effect, since
"radiative effects due only to aerosols" would imply both ari and aci. The notation for ERFcs,af is
inconsistent between eqn and text (comma or not). Eqns 4 & 5 sum to give not ERF, so where is
this missing 'surface albedo' term (ERF-ERFcsaf) and does it matter? It must certainly be counted
as an aerosol ERF. [OK, I see this in Table 3 later, is the cs,af just noise?]
*This section has been re-written and clarified, with additional discussion and explanation of the*
*terms and their meanings.*

L254ff- This is odd, you said just above the SARF is calculated from ERF – A_trop, so of course
this should give 0% difference. Are you just checking the math? Also, with SARF calculated as a
residual term, and the ability to denote and sum all the A_trop being highly uncertain, SARF
would not appear to be very certain. In fact the SARF term would depend on the models' ability
to diagnose A_trop correctly.

This has been explained more clearly in Section 4.1.2, and the reason for the difference in the
calculations clarified as due to the way the IRF is calculated in the kernel method. We have now
used the IRF instead of the SARF in this section, as it is more consistent with previous work.
L260- With BC, the long-standing problem is that some models get far too much in the
stratosphere and that would cause a very large SARF.
We have added more discussion of the BC results, and the reasons for model differences.
L285- in this figure and some others, please define carefully what the shorthand for the terms
means.
We have clarified the terms and included the definitions in the figure captions.
L296ff- This is a very good discussion of the aerosol components!
Thank you, nice to hear what is good about the paper!
L321- This would be a much better lead off to the aerosol section and analysis, begin with the big
picture before the weeds.
We have moved this to the top of the results section.
L343 – just put this table into the figure, it is just a summary of the bars anyway.
We have revised the plot, added error bars for both inter-annual variability and the multi-model
mean, and moved the relevant tables to the supplementary material.
L354- Please, stop wasting space, Table 5 and Fig 4 has the same information. If you want to
show AOD, then add it to the figure..
We have removed this plot and added the information to Table 4.
L370ff & Fig 5- I do not understand the purpose of this AOD scaling, it really does little to help.
The figure shows the key data: ERF from parts vs ERF from all. The ERF-parts consistently over
accounts for the combined ERF. This is as expected wince the cloud effects are largest in a pi
atmosphere with little background aerosols. So herein lies one of the fundamental problems with
the AerChemMIP that must be acknowledged and accounted for. I am not sure that scaling by
AOD is any more justified than just scaling the indirect to match – An interesting question is
whether the ERF-ari sum of the parts equals the whole? Keep this fudge factor simple since there
is no correct way to do this.
I have moved the discussion on scaling to the Supplementary material, and removed the
unnecessary figure. I have added appropriate caveats on the non-linearity and the effect of using
the PI atmosphere, and the effect on these results.
L393- Again, this title jars a bit, SO2 and NH3 are reactive gases.
This has been renamed to reactive greenhouse gases.
L406- How big is the error in GHGas ERF if one ignores clouds? I would think large.
We have added a discussion of the expected size of the cloud masking effect.
Fig 6 & Table 6, can easily combine and understand the Std Dev better.
This has been done, as for the previous ERF plots. The Fig. now contains the error bars and the
multi-model means, with a Table of values in the Supplemental material.

L419- Here is a case where some assessment is due as to how well these model simulations are
accurate in the sense of including all the effects. As noted the N2O-O3 link is important and
missing in some, and the other key link between N2O and tropospheric OH and methane (and
maybe aerosols) is established but missing here.
We have added more discussion on this. For the relevant impacts on stratospheric/tropospheric
ozone, and methane lifetime, only the models with appropriate chemical processes are
considered.
L431-441- This is a very interesting and important discussion about the additivity of the
components. I suspect that the CH4 result is similarly affected.
BTW, where is the effect of stratospheric H2o form CH4 noted or counted?
We have added plots of the water vapour changes from the CH4 experiment in the Supplementary
Materials.
L480- Yes halocarbons, but also N2O, and N2O may be more important since it depletes ozone in
the tropics (there are papers on this). I do not see what difference stratospheric Cl will make on
the total lifetime – hopefully the Stevenson paper becomes referencable. Also, you need to be
careful here since the methane feedback factors ff, apply to the PI atmosphere, and that is
different from the present, particularly lower CO and NOx…. . The feedback factor used for
GWPs etc, is the current one, not the PI one, so these results should NOT be used to change any
previous assessments.
We have mentioned the change due to N2O as well, and clarified that the f factors are starting
from a pre-industrial atmosphere.
Fig 11 is really hard to understand or see clearly, it will need a cleanup.
This figure has been revised to make it cleaner and easier to read.
L525- Here is a very important statement and I am not sure that you have put together all the
reasonable uncertainties or non-linear scaling. The individual components here must be adjusted
to recognize that the total ERF (with all simultaneously) is much less than the sum of the
components. Thus you cannot recommend the individual results without scaling and without
increasing the uncertainty.
We have revised the discussion to explain the differences between the individual components and
the total, and the implications for uncertainty.
L561- Same as above. The individual values will not sum correctly and so these do NOT reflect
the ERF of NOx emissions as we progress from 1850 to 2014. Thus they should not be used as
part of an assessment until they are more critically evaluated and put in context.
These are all very important AerChemMIP results, and the analysis here is highly valuable, but
their use in attribution and related studies should reflect the bias and uncertainties in combining
nonlinear parts that were calculated separately, and in basing these all on a pi-atmosphere.

This discussion has been revised to explain the differences between the sum of NOx and VOC
and the total. The ERFs are typically defined starting from a pre-industrial atmosphere, there is
no unique way to reflect the ERF of NOx emissions as we progress from 1850 to 2014
(subtracting components from a present atmosphere would overestimate the ERF).

**Effective radiative forcing from emissions of reactive gases and aerosols – a multi-model comparison**

Gillian D. Thornhill[1], William J. Collins[1], Ryan J. Kramer[2], Dirk Olivié[3], Ragnhild B. Skeie[4,] Fiona O'Connor[5,] Nathan L. Abraham[6], Ramiro Checa Garcia[7], Susanne E. Bauer[8], Makoto Deushi[9], Louisa K. Emmons[10], Piers Forster[11], Larry W. Horowitz[12], Ben Johnson[5], James Keeble[6], Jean-Francois Lamarque[10], Martine Michou[13], Mike Mills[10], Jane P. Mulcahy[5], Gunnar Myhre[4], Pierre Nabat[13], Vaishali Naik[12], Naga Oshima[9], Michael Schulz[3], Christopher J. Smith[11], Toshihiko Takemura[14], Simone Tilmes[10], Tongwen Wu[15], Guang Zeng[16], Jie Zhang[15].

[1]Department of Meteorology, University of Reading, Reading, RG6 6BB, UK

[2]Climate and Radiation Laboratory, NASA Goddard Space Flight Center, Greenbelt, MD 20771,USA, and Universities Space Research Association, 7178 Columbia Gateway Drive, Columbia, MD 21046, USA

[3]Norwegian Meteorological Institute, Oslo, Norway

[4] CICERO – Centre for International Climate and Environmental Research Oslo, Oslo, Norway

[5] Met Office, Exeter, UK

[6]Department of Chemistry, University of Cambridge, Lensfield Road, Cambridge, CB2 1EW, U.K., National Centre for Atmospheric Science, U.K

[7]IPSL/LSCE CEA-CNRS-UVSQ-UPSaclay UMR Gif sur Yvette, FRANCE

[8] NASA Goddard Institute for Space Studies, USA

[9] Meteorological Research Institute, Tsukuba, Japan

[10] National Center for Atmospheric Research, Boulder, CO, USA

[11] University of Leeds, Leeds, UK and International Institute for Applied Systems Analysis (IIASA), Laxenburg, Austria

[12] NOAA, Geophysical Fluid Dynamics Laboratory (GFDL), Princeton, NJ 08540-6649

[13] Centre National de Recherches Météorologiques, Meteo-France, Toulouse Cedex, France

[14] Research Institute for Applied Mechanics, Kyushu University, Japan

[15] Climate System Modeling Division, Beijing Climate Center, Beijing, China

[16] NIWA, Wellington, New Zealand

*Correspondence to*: Gillian D. Thornhill (g.thornhill@reading.ac.uk)

**Abstract**

This paper quantifies effective radiative forcing (ERF) of the present-day anthropogenic emissions of $NO_X$, VOCs (including CO), $SO_2$, $NH_3$, black carbon and organic carbon. Effective radiative forcings from pre-industrial to present-day changes in the concentrations of methane, $N_2O$ and ozone-depleting halocarbons are quantified. Concentration and emission changes of reactive species can cause multiple changes in the composition of radiatively active species: tropospheric ozone, stratospheric ozone, stratospheric water vapour, secondary inorganic and organic aerosol and methane. Where possible we break down the ERFs from each emitted species into the contributions from the composition changes.

The ERFs are calculated using models that participated in the AerChemMIP experiments as part of the CMIP6

project.

The 1850 to 2014 multi-model mean ERFs (± standard deviations) are -1.03 ± 0.37 Wm$^{-2}$ for SO$_2$ emissions, -

0.25 ± 0.09 Wm$^{-2}$ for organic carbon (OC), 0.15 ± 0.17 Wm$^{-2}$ for black carbon (BC), for NH$_3$ it is -0.07 ± 0.01Wm$^{-2}$

[revised manuscript text omitted]

Aerosol scheme is coupled to the tropospheric chemistry scheme which includes inorganic chemistry of $O_x$, $NO_x$, $HO_x$, CO, and organic chemistry of $CH_4$ and higher hydrocarbons using the CBM4 scheme and the stratospheric chemistry scheme which includes chlorine and bromine chemistry together with polar stratospheric clouds. | (Bauer et al., 2020;Shindell et al., 2001;Shindell et al., 2003;Gery et al., 1989;Shindell et al., 2006) |

| CESM2-WACCM | 0.9 (lat) x 1.25 (lon), 70 levels | Chemistry and aerosols for the troposphere, stratosphere, mesosphere and lower thermosphere are calculated interactively.  It simulates 228 compounds, including the 4-mode Modal Aerosol Model (MAM4).  This version of MAM4 is modified to allow for the simulation of stratospheric aerosols from volcanic eruptions (from their $SO_2$ emissions) and oxidation of OCS. The representation of secondary organic aerosols follows the Volatility Basis Set approach. | (Emmons et al., 2020;Danabasoghu, 2019;Danabasoglu, 2019;Gettelman et al., 2019;Tilmes et al., 2019) Mills et al., 2016) |
|---|---|---|---|

(Table for European models updated from:

Crescendo Report Horizon 2020

H2020-SC5-2014 Advanced Earth-system models (Grant Agreement 641816)

Coordinated Research in Earth Systems and Climate: Experiments, kNowledge, Dissemination and
OutreachDeliverable D_6.2

**S2 Tables of ERF and ERF_ts for all models analysed**

By removing the adjustment due to the changes in the land surface temperature (as calculated from radiative
kernels) we show the ERF_Ts for those models where the adjustment was available in the following table.

The tables below give  the 1850-2014 ERF and the ERF_Ts calculated from the TOA flux differences for each
model for each experiment.

**Table S 2 ERFs and ERF_ts for the aerosols, including multi-model means with standard errors.**

| ERF | aer | | BC | | OC | | SO2 | | NH3 | |
|---|---|---|---|---|---|---|---|---|---|---|
| Wm$^{-2}$ | ERF | ERF_ts | ERF | ERF_ts | ERF | ERF_ts | ERF | ERF_ts | ERF | ERF_ts |
| CNRM-ESM2 | -0.74 | -0.79 | 0.11 | 0.11 | -0.17 | -0.18 | -0.75 | -0.78 | - | - |
| UKESM1 | -1.10 | -1.15 | 0.37 | 0.36 | -0.21 | -0.23 | -1.36 | -1.41 | - | - |
| MRI-ESM2 | -1.21 | -1.24 | 0.25 | 0.27 | -0.32 | -0.32 | -1.37 | -1.42 | - | - |
| BCC-ESM1 | -1.47 | -1.54 | 0.21 | 0.21 | - | - | -1.54 | -1.62 | - | - |

| | | | | | | | | | |
|---|---|---|---|---|---|---|---|---|---|
| **MIROC6** | -1.01 | -1.07 | -0.21 | -0.24 | -0.23 | -0.26 | -0.64 | -0.67 | - | - |
| **NorESM2** | -1.21 | -1.21 | 0.30 | 0.30 | -0.22 | -0.23 | -1.28 | -1.29 | - | - |
| **GFDL-ESM4** | -0.70 | -0.73 | - | - | - | - | - | - | - | - |
| **GISS-E2-1** | -0.90 | -0.95 | 0.06 | 0.06 | -0.44 | -0.45 | -0.62 | -0.65 | -0.08 | - |
| **IPSL-INCA** | -0.75 | - | 0.10 | - | -0.15 | - | -0.69 | - | -0.06 | - |
| **MultiModel Mean** | -1.01 | -1.09 | 0.15 | 0.16 | -0.25 | -0.28 | -1.03 | -1.12 | -0.07 | - |
| **S.D.** | 0.25 | 0.25 | 0.17 | 0.19 | 0.09 | 0.09 | 0.37 | 0.38 | 0.01 | |

**Table S 3 ERF, ERF_ts, multimodel means and standard error for the chemically reactive gases.**

| **ERF** | **CH4** | | **HC** | | **N2O** | | **NTCF** | | **O3** | | **NOx** | | **VOC** | |
|---|---|---|---|---|---|---|---|---|---|---|---|---|---|---|
| **Wm$^{-2}$** | **ERF** | **ERF_ts** | **ERF** | **ERF_ts** | **ERF** | **ERF_ts** | **ERF** | **ERF_ts** | **ERF** | **ERF_ts** | **ERF** | **ERF_ts** | **ERF** | **ERF_ts** |
| **CNRM-ESM2** | 0.44 | 0.46 | -0.10 | -0.10 | 0.32 | 0.33 | -0.74 | -0.79 | - | - | - | - | - | - |
| **UKESM1** | 0.97 | 1.00 | -0.18 | -0.19 | 0.25 | - | -1.03 | -1.03 | 0.21 | 0.22 | 0.03 | 0.04 | 0.33 | 0.34 |
| **MRI-ESM2** | 0.70 | 0.73 | 0.31 | 0.31 | 0.19 | - | -1.08 | - | 0.06 | 0.15 | -0.02 | -0.02 | -0.03 | -0.03 |
| **BCC-ESM1** | 0.68 | 0.72 | - | - | - | - | | | 0.21 | 0.24 | 0.12 | 0.15 | -0.04 | -0.04 |
| **MIROC6** | - | - | - | - | - | - | -0.85 | -0.83 | - | - | - | - | - | - |
| **NorESM2** | 0.37 | 0.39 | - | - | 0.23 | 0.24 | - | - | - | - | - | - | - | - |
| **GFDL-ESM4** | 0.68 | 0.70 | 0.06 | 0.08 | - | - | -0.51 | -0.55 | 0.27 | 0.29 | 0.14 | 0.16 | 0.08 | -0.08 |
| **GISS-E2-1** | 0.78 | 0.80 | 0.28 | 0.29 | 0.20 | 0.20 | -0.92 | -0.98 | 0.23 | 0.23 | 0.16 | 0.16 | 0.22 | 0.22 |
| **CESM2-WACCM** | 0.72 | 0.76 | 0.34 | 0.38 | 0.39 | 0.40 | -0.89 | -0.89 | - | - | 0.40 | 0.44 | 0.00 | 0.00 |
| **IPSL-INCA** | - | - | - | - | - | - | - | - | - | - | - | - | - | - |
| **MultiModel Mean** | 0.67 | 0.69 | 0.12 | 0.13 | 0.26 | 0.29 | -0.86 | -0.85 | 0.20 | 0.23 | 0.14 | 0.15 | 0.09 | 0.07 |
| **S. D.** | 0.17 | 0.18 | 0.21 | 0.22 | 0.07 | 0.08 | 0.18 | 0.15 | 0.07 | 0.05 | 0.13 | 0.14 | 0.14 | 0.15 |

**In Table S4 the mean ERF per emissions or concentrations is given for each experiment.**

**Table S 4 Table of ERF/emissions or concentrations. Emissions for NOx are scaled to Tg of NO2**

| ERF/emission or concentration | BC (Wm$^{-2}$/Tg) | SO2 (Wm$^{-2}$/Tg) | OC (Wm$^{-2}$/Tg) | NH3 (Wm$^{-2}$/Tg) | NOx (scaled to Wm$^{-2}$/Tg NO2) | CH4 (Wm$^{-2}$/ppb) | HC (Wm$^{-2}$/ppb) | N2O (Wm$^{-2}$/ppb) |
|---|---|---|---|---|---|---|---|---|
| | | | | | | | | |

| | | | | | | | |
|---|---|---|---|---|---|---|---|
| 0.0212 | -0.0094 | -0.0147 | -0.0013 | 0.0010 | 0.0007 | 0.1200 | 0.0048 |

**S3 Kernel Breakdown of atmospheric adjustments for each experiment**

The full breakdowns of the rapid adjustments as calculated from the kernels is shown for each of the models and experiments where the relevant data was available and shows the differences in models for how the rapid adjustments from different processes contributed to the overall rapid adjustment.

**Table S5a Adjustments for piClim-aer experiment**

| piClim-aer | CNRM-ESM2 | UKESM1 | MRI-ESM2 | BCC-ESM1 | MIROC6 | NorESM2 | GFDL-ESM4 | GISS-E2-1 |
|---|---|---|---|---|---|---|---|---|
| albedo | -0.017 | -0.049 | -0.009 | -0.095 | -0.026 | -0.015 | -0.044 | 0.003 |
| cloud | -0.661 | -0.915 | -0.842 | -0.900 | -0.945 | -1.093 | -0.452 | -0.581 |
| W.V. | -0.055 | 0.017 | 0.169 | -0.008 | -0.085 | 0.029 | 0.094 | -0.046 |
| T_trop | 0.107 | 0.023 | -0.243 | 0.092 | 0.183 | 0.038 | -0.138 | 0.137 |
| T_strat | -0.015 | -0.006 | -0.014 | -0.038 | 0.010 | -0.054 | -0.027 | -0.038 |
| T_surface | 0.054 | 0.049 | 0.032 | 0.075 | 0.059 | 0.001 | 0.035 | 0.052 |

**Table S5b Adjustments for piClim-BC experiment**

| piClim-BC | CNRM-ESM2 | UKESM1 | MRI-ESM2 | BCC-ESM1 | MIROC6 | NorESM2 | GISS-E2-1 |
|---|---|---|---|---|---|---|---|
| albedo | 0.003 | -0.013 | 0.067 | 0.015 | -0.002 | 0.076 | 0.046 |
| cloud | -0.010 | -0.013 | 0.163 | 0.053 | -0.282 | -0.067 | -0.184 |
| W.V. | 0.060 | 0.057 | 0.329 | 0.076 | -0.042 | 0.097 | 0.038 |
| T_trop | -0.088 | -0.137 | -0.509 | -0.131 | 0.033 | -0.137 | -0.051 |
| T_strat | -0.003 | 0.043 | 0.021 | -0.004 | -0.008 | -0.025 | -0.003 |
| T_surface | 0.001 | 0.005 | -0.022 | -0.002 | 0.021 | -0.003 | 0.003 |

**Table S5c Adjustments for piClim-OC experiment**

| piClim-OC | CNRM-ESM2 | UKESM | MRI | MIROC6 | NorESM2 | GISS-E2-1 |
|---|---|---|---|---|---|---|
| albedo | 0.008 | -0.015 | -0.006 | -0.017 | 0.002 | 0.001 |
| cloud | -0.083 | 0.052 | -0.129 | -0.087 | -0.100 | -0.290 |
| W.V. | 0.000 | -0.018 | -0.009 | -0.004 | -0.008 | -0.054 |
| T_trop | 0.025 | 0.023 | -0.016 | 0.029 | 0.066 | 0.057 |
| T_strat | -0.019 | -0.003 | -0.010 | -0.008 | -0.016 | -0.015 |
| T_surface | 0.011 | 0.015 | 0.009 | 0.035 | 0.010 | 0.012 |

**Table S5d Adjustments for piClim-SO2 experiment**

| piClim-SO2 | CNRM-ESM2 | UKESM1 | MRI-ESM2 | BCC-ESM1 | MIROC6 | NorESM2 | GISS-E2-1 |
|---|---|---|---|---|---|---|---|
| albedo | 0.01 | -0.03 | -0.04 | -0.10 | -0.02 | -0.09 | -0.04 |
| cloud | -0.49 | -0.79 | -0.73 | -0.43 | -0.40 | -0.96 | -0.02 |
| W.V. | -0.04 | -0.10 | -0.05 | -0.07 | -0.06 | -0.05 | -0.06 |
| T_trop | 0.10 | 0.20 | 0.08 | 0.22 | 0.11 | 0.16 | 0.14 |
| T_strat | -0.01 | -0.02 | -0.02 | -0.03 | 0.00 | -0.03 | -0.01 |
| T_surface | 0.03 | 0.04 | 0.06 | 0.07 | 0.04 | 0.01 | 0.03 |

**Table S5e Adjustments for piClim-CH4 experiment**

| piClim-CH4 | CNRM-ESM2 | UKESM1 | MRI-ESM2 | BCC-ESM1 | NorESM2 | GFDL-ESM4 | GISS-E2-1 | CESM2-WACCM |
|---|---|---|---|---|---|---|---|---|
| albedo | 0.019 | 0.019 | 0.013 | 0.031 | 0.010 | 0.014 | 0.007 | 0.023 |
| cloud | -0.038 | 0.242 | -0.056 | 0.008 | -0.041 | -0.054 | 0.035 | 0.045 |
| W.V. | 0.071 | 0.109 | 0.070 | 0.055 | 0.018 | 0.117 | 0.021 | 0.068 |
| T_trop | -0.084 | -0.162 | -0.138 | -0.127 | -0.080 | -0.147 | -0.056 | -0.082 |
| T_strat | 0.114 | 0.115 | 0.124 | 0.039 | 0.057 | 0.109 | 0.053 | 0.103 |
| T_surface | -0.017 | -0.031 | -0.028 | -0.046 | -0.018 | -0.023 | -0.019 | -0.031 |

**Table S5f Adjustments for piClim-HC experiment**

| piClim-HC | CNRM-ESM2 | UKESM1 | MRI-ESM2 | GFDL-ESM4 | GISS-E2-1 | CESM2-WACCM |
|---|---|---|---|---|---|---|
| albedo | 0.01 | -0.02 | 0.00 | 0.00 | -0.01 | 0.01 |
| cloud | -0.05 | -0.09 | -0.02 | -0.03 | 0.03 | 0.06 |
| W.V. | -0.02 | -0.09 | 0.02 | -0.01 | -0.01 | 0.01 |
| T_trop | 0.03 | 0.10 | -0.03 | -0.03 | -0.01 | 0.00 |
| T_strat | 0.21 | 0.47 | 0.16 | 0.26 | 0.30 | 0.16 |
| T_surface | 0.00 | 0.01 | 0.00 | -0.02 | -0.01 | -0.03 |

**Table S5g Adjustments for piClim-N2O experiment**

| piClim-NO2 | CNRM-ESM2 | UKESM1 | MRI-ESM2 | NorESM2 | GISS-E2-1 | CESM2-WACCM |
|---|---|---|---|---|---|---|
| albedo | 0.021 | 0.001 | 0.005 | 0.003 | 0.000 | 0.008 |
| cloud | -0.010 | 0.047 | 0.004 | 0.059 | 0.026 | 0.119 |
| W.V. | 0.038 | -0.002 | -0.015 | -0.012 | -0.006 | 0.016 |
| T_trop | -0.035 | -0.006 | -0.022 | -0.037 | -0.002 | -0.014 |

| | | | | | | |
|---|---|---|---|---|---|---|
| T_strat | 0.094 | 0.074 | 0.108 | 0.003 | 0.130 | 0.089 |
| T_surface | -0.016 | -0.011 | -0.005 | -0.014 | 0.005 | -0.016 |

**Table S5h Adjustments for piClim-NOx experiment**

| piClim-NOx | UKESM1 | MRI-ESM2 | BCC-ESM1 | GFDL-ESM4 | GISS-E2-1 | CESM2-WACCM |
|---|---|---|---|---|---|---|
| albedo | -0.005 | 0.018 | 0.023 | 0.010 | 0.003 | 0.041 |
| cloud | -0.036 | -0.041 | 0.007 | -0.065 | -0.052 | 0.104 |
| Spec. Hum. | -0.003 | -0.040 | 0.031 | 0.029 | 0.005 | 0.062 |
| T_trop | 0.002 | -0.014 | -0.063 | -0.057 | -0.001 | -0.056 |
| T_strat | -0.024 | -0.161 | 0.029 | 0.038 | 0.056 | 0.095 |
| T_surface | -0.012 | -0.012 | -0.026 | -0.025 | -0.001 | -0.030 |

**Table S5i Adjustments for piClim-O3 experiment**

| piClim-O3 | UKESM1 | MRI-ESM2 | BCC-ESM1 | GFDL-ESM4 | GISS-E2-1 |
|---|---|---|---|---|---|
| albedo | 0.001 | 0.002 | 0.026 | 0.009 | -0.003 |
| cloud | 0.091 | -0.126 | 0.096 | -0.079 | -0.021 |
| W.V. | 0.006 | 0.010 | 0.054 | 0.082 | -0.004 |
| T_trop | -0.034 | -0.062 | -0.115 | -0.083 | -0.021 |
| T_strat | 0.009 | -0.036 | 0.047 | 0.094 | 0.113 |
| T_surface | -0.016 | -0.010 | -0.037 | -0.015 | 0.002 |

**Table S5j Adjustments for piClim-VOC experiment**

| piClim-VOC | UKESM1 | MRI-ESM2 | BCC-ESM1 | GFDL-ESM4 | GISS-E2-1 | CESM2-WACCM |
|---|---|---|---|---|---|---|
| albedo | 0.008 | 0.006 | 0.000 | 0.002 | -0.001 | 0.004 |
| cloud | 0.190 | -0.147 | -0.020 | -0.004 | -0.001 | 0.033 |
| W.V. | 0.023 | 0.050 | 0.022 | 0.067 | 0.010 | -0.049 |
| T_trop | -0.009 | -0.072 | -0.046 | -0.059 | -0.006 | 0.072 |
| T_strat | 0.042 | 0.054 | -0.001 | 0.024 | 0.064 | 0.021 |
| T_surface | -0.006 | -0.016 | -0.011 | -0.011 | 0.002 | 0.000 |

Bar charts showing the atmospheric adjustments calculated from the kernel analysis are included below,
showing adjustments for surface albedo, cloud, water vapour, tropospheric temperature, stratospheric
temperature, and surface temperature.

[Figure]

[Figure]

[Figure]

[Figure]

[Figure]

[Figure]

[Figure]

[Figure]

[Figure]

[Figure]

**Figure S 1 Plots showing the breakdown of rapid adjustments for all experiments and models with the appropriate**
**diagnostics**

**Table S 5 Comparison of IRF and cloud adjustment with Smith et al. (2020) for piClim-aer experiment.**

| piClim-aer (Wm⁻²) | IRF (This work) | IRF (Smith et al., 2020)) | IRF Diff | Cloud adj. (This work) | Cloud adj. (Smith et al., 2020) | Cloud adj. Diff | IRF+Cld Adj (this work) | IRF+Cloud Adj. (Smith et al., 2020) | Total Diff | % Diff |
|---|---|---|---|---|---|---|---|---|---|---|
| CNRM-ESM2-1 | -0.15 | -0.75 | 0.60 | -0.66 | -0.06 | -0.60 | -0.82 | -0.81 | -0.01 | 0.63 |
| GFDL-ESM4 | -0.16 | -0.37 | 0.21 | -0.45 | -0.26 | -0.19 | -0.61 | -0.63 | 0.02 | -2.77 |
| GISS-E2-1-G (p3) | -0.42 | -1.00 | 0.58 | -0.60 | -0.01 | -0.59 | -1.02 | -1.01 | -0.01 | 1.33 |
| MIROC6 | -0.21 | -1.13 | 0.92 | -0.95 | -0.02 | -0.93 | -1.16 | -1.15 | -0.01 | 0.80 |
| MRI-ESM2-0 | -0.30 | -0.46 | 0.16 | -0.85 | -0.68 | -0.17 | -1.15 | -1.14 | -0.01 | 0.79 |
| NorESM2-LM | -0.11 | -1.09 | 0.98 | -1.11 | -0.08 | -1.03 | -1.22 | -1.17 | -0.05 | 4.28 |
| UKESM1-0-LL | -0.22 | -0.97 | 0.75 | -0.93 | -0.18 | -0.75 | -1.15 | -1.15 | 0.00 | 0.17 |
| **Mean** | **-0.23** | **-0.82** | **0.60** | **-0.79** | **-0.18** | **-0.61** | **-1.02** | **-1.01** | **-0.01** | **1.01** |

(a)

[Figure]

[Figure]

(b)

**Figure S 2 Scatter plots comparing direct and indirect aerosol effects from Ghan diagnostics (x-axes) with**
**kernel-derived breakdown (y-axes). Points are from 4 models that included Ghan diagnostics (CNRM-**
**ESM1, UKESM1, MRI-ESM2, NorESM2) (a) Comparison of IRFari with kernel IRF. (b) Comparison of**
**ERFaci with kernel cloud adjustment (Acloud).**

**S4 AOD scaling**

[Figure]

In Fig (S2) we compare the ERF originally calculated from the radiative fluxes for the (piClim-xx –piClim-
control) experiments – referred to as Calc ERF to the ERF contributions obtained from using the AOD scaling,

**Figure S 3 Comparison of the ERF calculated from radiative fluxes with that from the ERF from AOD-scaled values.**

e.g. the BC AOD in the piClim-BC experiment. In general, the change in the single species is responsible for
most of the change in the ERF in these experiments, however in the MIROC6 piClim-OC experiment there is a
significant contribution from the organic carbon, indicating this is not as clean a method for obtaining the
scaling in this case as for the other models and experiments. In the case of NorESM2 for the $SO_2$ experiment we
also have some contribution from the OA, which may be attributable to the way the nucleation scheme works in
NorESM2. Their nucleation scheme looks at the combination of H2SO4 and low-volatile organic vapours
(precursors of SOA), so changing the SO2 emissions might therefore indirectly change the pathway for the SOA
precursors, leading to a shift in how much nucleates and how much condensates.  This might lead to a difference
in lifetime of SOA (which is part of OM), leading to differences in the OM burden or AOD. (Dirk Olive, pers.
Communication).

In Table S6 the ERF per Tg burden is shown for the piClim-BC, piClim-SO2 and piClim-OC experiments.

    **Table S 6 Table of ERF/burden for individual aerosol experiments**

| ERF/burden (Wm$^{-2}$ Tg$^{-1}$) | CNRM-ESM2 | MIROC6 | NorESM2 | UKESM1 | GISS-E2-1 | MRI-ESM2 | BCC-ESM1 | IPSL-INCA |
|---|---|---|---|---|---|---|---|---|
| piClim-BC | 1.43 | -2.49 | 2.38 | 4.07 | 0.92 | 1.74 | 1.63 | 0.90 |
| piClim-OC | -0.68 | -0.67 | -0.55 | -0.45 | -1.42 | -1.02 | | -0.35 |
| piClim-SO2 | -1.12 | -0.93 | -1.17 | -1.34 | -1.01 | -1.47 | -1.33 | -0.64 |

    ## S5 Detailed plot of the atmospheric adjustments for the piClim-CH4 model results

    The rapid adjustments for the CH4 experiment are broken down to show the model differences and the
    contributions of the individual rapid adjustments to the overall rapid adjustment contribution to the ERF.

[Figure]

**Figure S 4 Plots showing the rapid adjustments for the piClim-CH4 experiments**

    ## S5 Plots of the Ghan Calculations

 We also plotted the breakdown of the ERF into the ERFari, ERFcloud and the ERFcs,af (clear sky, no aerosol)
 for models with the appropriate diagnostics, shown in Fig. S4 below.

[Figure]

**Figure S 5 Breakdown of the ERFs using the double-call method, showing IRFari, ERFcs,af and ERFaci**

**S6 Experiments using NorESM2 to examine adding 2014 aerosols to an atmosphere with 2014 oxidants.**

The following sensitivity experiments were done with the NorESM aerosol scheme. To study the effect of $SO_2$ emissions, we have done a few extra simulations in addition to piClim-control and piClim-SO2 : two additional experiments which we called piClim-oxid and piClim-oxidSO2.

These experiments are :

(1) piClim-control : $SO_2$ emissions are 1850, oxidants are 1850

(2) piClim-$SO_2$ : $SO_2$ emissions are 2014, oxidants are 1850

(3) piClim-oxid : $SO_2$ emissions are 1850, oxidants are 2014

(4) piClim-oxid+$SO_2$ : $SO_2$ emissions are 2014, oxidants are 2014

The standard results in this paper compare (2) with (1) : this gives ERF = -1.303 W/m$^2$ (N.B the calculations here were done over 25 years, not 30 years as in the rest of the paper). It reflects the impact of adding $SO_2$ emissions in a clean pre-industrial atmosphere (both (1) and (2) have the oxidants on 1850 levels, as if NOx, CO, VOC, ... emissions are all 1850).

However, if we compare (4) with (3) : this gives -1.479 W/m$^2$. It is the impact of adding $SO_2$ emissions, already in a polluted atmosphere where NOx, CO, VOC, ... emissions are at 2014 levels, and therefore high oxidant values.

It shows that we have differences of the order of 13% : ERF = -1.303 W/m$^2$ compared to -1.479 W/m2.

Similar experiments for all aerosols together result in the following :

(1) piClim-control : aerosol emissions are 1850, oxidants are 1850

(2) piClim-aer : aerosol emissions are 2014, oxidants are 1850

(3) piClim-oxid : aerosol emissions are 1850, oxidants are 2014

(4) piClim-oxid+aer : aerosol emissions are 2014, oxidants are 2014

Comparing here (2) with (1) gives ERF = -1.214 W/m$^2$ and comparing (4) with (3) gives ERF = -1.458 W/m$^2$. This gives a difference of around 20%.

This result is only obtained in a simplified setup (prescribed oxidants), but it might give an indication of how the "chemical climate" affects the result.

The climate conditions (different temperature and deposition rates in 1850 and 2014) are of course not covered by the above experiment. It remains in an 1850 climate.

Finally, the impact of large emission reductions (like 100% for $SO_2$) can show a different sensitivity than smaller mitigation-type reduction sizes due to non-linearity.

(D. Olivie, pers. Comm).

**S7 Breakdown of Ozone changes**

Table S 7 Column ozone, and ozone changes resulting from changes concentrations (CH4, N2O, HC) or emissions (NOX, VOC, O3, NTCF) of reactive gases. The multi
model mean does not include the results for CNRM-ESM2 for tropospheric ozone.

| Experiment | CNRM-ESM2 | | UKSM1 | | MRI-ESM2 | | BCC-ESM1 | | GFDL-ESM4 | | GISS-E2 | | CESM2-WACCM | | Multi-model | |
|---|---|---|---|---|---|---|---|---|---|---|---|---|---|---|---|---|
| | trop | strat | trop | strat | trop | Strat | trop | strat | trop | strat | trop | strat | trop | strat | trop | strat |
| Control DU | | 303.0 ±0.2 | 25.71 ±0.06 | 313.2 ±0.6 | 19.88 ±0.04 | 294.8 ±0.4 | 23.20 ±0.03 | | 20.15 ±0.02 | 267.0 ±0.2 | 20.45 ±0.04 | 258.5 ±0.1 | 20.33 ±0.04 | 260.3 ±0.2 | 22.3 ±2.60 | 283 ±20 |
| CH4 DU | | +6.1±0.3 | 3.02 ±0.08 | +2.0 ±0.6 | +2.48 ±0.04 | +2.9 ±0.5 | +2.42 ±0.03 | | 2.50 ±0.04 | +2.1 ±0.2 | 2.17 ±0.05 | +5.3 ±0.2 | +3.15 ±0.04 | +2.9±0.2 | +2.6 ±0.3 | +4 ±2 |
| NOx DU | | | 5.20 ±0.08 | +4.6 ±0.6 | +4.09 ±0.05 | +10.5 ±0.5 | 7.23 ±0.03 | | 6.61 ±0.03 | +1.1 ±0.2 | 9.19 ±0.05 | +1.0 ±0.2 | +6.97 ±0.04 | +0.1 ±0.2 | +6.5 ±1.6 | +3 ±3 |
| VOC DU | | | 1.47 ±0.08 | +1.6 ±0.6 | +1.99 ±0.05 | +2.0 ±0.5 | 0.79 ±0.03 | | 1.94 ±0.03 | +2.5 ±0.2 | 1.90 ±0.05 | -1.9 ±0.3 | +1.57 ±0.05 | +2.0 ±0.2 | +1.6 ±0.4 | +1 ±1 |
| O3 DU | | | 6.86 ±0.08 | +5.1 ±0.6 | +7.51 ±0.04 | 7.2 ±0.5 | 8.52 ±0.03 | | 9.46 ±0.03 | +2.8 ±0.2 | 11.38 ±0.06 | -0.6 ±0.3 | | | +8.7 ±1.6 | +4 ±3 |
| N2O DU | | -6.7 ±0.3 | 0.16 ± 0.08 | -3.1 ±0.6 | --0.05 ±0.04 | -4.7 ±0.5 | | | | | 0.23 ±0.05 | -7.6 ±0.2 | +0.41 ±0.04 | -4.5 ±0.2 | +0.2 ±0.2 | -5 ±1 |
| HC DU | | -23.4 ±0.8 | -2.12± 0.08 | -38.2 ±0.6 | -0.41 ±0.05 | -13.4 ±0.5 | | | -1.51 ±0.0 | -23.3 ±0.2 | -2.54 ±0.05 | -24.2 ±0.2 | -0.61 ±0.06 | -22.7 ±0.4 | -1.4 ±0.8 | -23 ±8 |

**Table S 8 Percentage change in total aerosol mass (sulphate, nitrate and secondary organic) from the reactive gas experiments.**

| Experiment | UKESM | MRI-ESM2 | BCC-ESM2 | GFDL-ESM4 | | | GISS-E2 | | | CESM2-WACCM | |
|---|---|---|---|---|---|---|---|---|---|---|---|
| | SO4 | SO4 | SO4 | SO4 | NO3 | SOA | SO4 | NO3 | SOA | SO4 | SOA |
| CH4 | -2 | +2 | +1 | +6 | 0 | 0 | -1 | -11 | +1 | -1 | -1 |
| NOx | -2 | 0 | +9 | -8 | +200 | +1 | -19 | +120 | +19 | +2 | -10 |
| VOC | -3 | +1 | 0 | +4 | -5 | +8 | -2 | 0 | +1 | -2 | +34 |
| O3 | -4 | +1 | +10 | -3 | +190 | +7 | -21 | +130 | +21 | | |
| N2O | +1 | +1 | | | | | +1 | +7 | +1 | +1 | +1 |
| HC | +2 | +1 | | +1 | 0 | +1 | +1 | -11 | +1 | -1 | +1 |

**S8 Methane Lifetime**

**Table S 9 Methane lifetime (years), and change due to each experiment (%). Multi-model mean and standard deviation. Lifetimes assume a soil loss of 120 years. Stratospheric loss is included in the model calculations.**

| Experiment | UKESM1 | CESM2-WACCM | GFDL-ESM4 | BCC | GISS-E2 | MRI-ESM2 | Multi-model |
|---|---|---|---|---|---|---|---|
| Control years | 8.0 | 8.7 | 9.6 | 6.3 | 13.4 | 10.1 | 10.0 ±1.9 |
| CH4 % | +22 | +22 | +21 | +26 | +18 | +22 | 22±3 |
| NOx % | -25 | -35 | -33 | | -46 | -26 | -33±8 |
| VOC % | +11 | | +15 | | +27 | +21 | +19±6 |
| O3 % | -19 | | -24 | | -40 | -20 | -16±9 |
| HC % | -4.9 | | -7.5 | | -0.6 | -2.4 | -3.7 ±2.4 |
| N2O | -1.2 | -2.8 | | | -3.9 | -1.3 | -2.0 ±1.1 |

[Figure]

**Figure S 6 Ozone column values for the troposphere and stratosphere for the reactive greenhouse gas experiments**

[Figure]

**Figure S 7 Percentage changes in water vapour from the piClim-CH4 experiments.**

Abdul-Razzak, H., and Ghan, S. J.: A parameterization of aerosol activation: 2. Multiple aerosol types, Journal of Geophysical Research: Atmospheres, 105, 6837-6844, 10.1029/1999jd901161, 2000.

Archibald, A. T., O'Connor, F. M., Abraham, N. L., Archer-Nicholls, S., Chipperfield, M. P., Dalvi, M., Folberth, G. A., Dennison, F., Dhomse, S. S., Griffiths, P. T., Hardacre, C., Hewitt, A. J., Hill, R.,

Johnson, C. E., Keeble, J., Köhler, M. O., Morgenstern, O., Mulchay, J. P., Ordóñez, C., Pope, R. J., Rumbold, S., Russo, M. R., Savage, N., Sellar, A., Stringer, M., Turnock, S., Wild, O., and Zeng, G.: Description and evaluation of the UKCA stratosphere-troposphere chemistry scheme (StratTrop vn 1.0) implemented in UKESM1, Geosci. Model Dev. Discuss., 2019, 1-82, 10.5194/gmd-2019-246, 2020.

Balkanski, Y., Myhre, G., Gauss, M., Rädel, G., Highwood, E. J., and Shine, K. P.: Direct radiative effect of aerosols emitted by transport: from road, shipping and aviation, Atmos. Chem. Phys., 10, 4477-4489, 10.5194/acp-10-4477-2010, 2010.

Bauer, S. E., Tsigaridis, K., Faluvegi, G., Kelley, M., Lo, K. K., Miller, R. L., Nazarenko, L., Schmidt, G. A., and Wu, J.: Historical (1850–2014) Aerosol Evolution and Role on Climate Forcing Using the GISS ModelE2.1 Contribution to CMIP6, Journal of Advances in Modeling Earth Systems, 12, e2019MS001978, 10.1029/2019ms001978, 2020.

Checa-Garcia, R., Hegglin, M. I., Kinnison, D., Plummer, D. A., and Shine, K. P.: Historical Tropospheric and Stratospheric Ozone Radiative Forcing Using the CMIP6 Database, Geophysical Research Letters, 45, 3264-3273, 10.1002/2017gl076770, 2018.

Danabasoghu, G.: NCAR CESM2-WACCM model output prepared for CMIP6 CMIP historical, https://doi.org/10.22033/ESGF/CMIP6.10071, https://doi.org/10.22033/ESGF/CMIP6.10071, 2019.

Danabasoglu, G.: NCAR CESM2-WACCM model output prepared for CMIP6 ScenarioMIP, https://doi.org/10.22033/ESGF/CMIP6.10026, http://cera-www.dkrz.de/WDCC/meta/CMIP6/CMIP6.ScenarioMIP.NCAR.CESM2-WACCM, 2019.

Di Biagio, C., Formenti, P., Balkanski, Y., Caponi, L., Cazaunau, M., Pangui, E., Journet, E., Nowak, S., Andreae, M., Kandler, K., Saeed, T., Piketh, S., Seibert, D., Williams, E., and Doussin, J.-F.: Complex refractive indices and single scattering albedo of global dust aerosols in the shortwave spectrum and relationship to iron content and size, Atmospheric Chemistry and Physics Discussions, 1-42, 10.5194/acp-2019-145, 2019.

Dunne, J. P., Horowitz, L. W., Adcroft, A. J., Ginoux, P., Held, I. M., John, J. G., Krasting, J. P., Malyshev, S., Naik, V., Paulot, F., Shevliakova, E., Stock, C. A., Zadeh, N., Balaji, V., Blanton, C., Dunne, K. A., Dupuis, C., Durachta, J., Dussin, R., Gauthier, P. P. G., Griffies, S. M., Guo, H., Hallberg, R. W., Harrison, M., He, J., Hurlin, W., McHugh, C., Menzel, R., Milly, P. C. D., Nikonov, S., Paynter, D. J., Ploshay, J., Radhakrishnan, A., Rand, K., Reichl, B. G., Robinson, T., Schwarzkopf, D. M., Sentman, L. T., Underwood, S., Vahlenkamp, H., Winton, M., Wittenberg, A. T., Wyman, B., Zeng, Y., and Zhao, M.: The GFDL Earth System Model version 4.1 (GFDL-ESM 4.1): Overall coupled model description and simulation characteristics, Journal of Advances in Modeling Earth Systems, n/a, e2019MS002015, 10.1029/2019ms002015, 2020.

Emmons, L. K., Schwantes, R. H., Orlando, J. J., Tyndall, G., Kinnison, D., -F., L. J., Marsh, D., Mills, M., Tilmes, S., Bardeen, C., Buchholz, R. R., Conley, A., Gettelman, A., Garcia, R., Simpson, I., Blake, D. R., Meinardi, S., and Pétron, G.: The Chemistry Mechanism in the Community Earth System Model version 2 (CESM2), J. Advances in Modeling Earth Systems, 12, https://doi.org/10.1029/2019MS001882, 2020.

Gery, M. W., Whitten, G. Z., Killus, J. P., and Dodge, M. C.: A photochemical kinetics mechanism for urban and regional scale computer modeling., Journal of Geophysical Research, 94, 925-956, 1989.

Gettelman, A., Mills, M. J., Kinnison, D. E., Garcia, R. R., Smith, A. K., Marsh, D. R., Tilmes, S., Vitt, F., Bardeen, C. G., McInerny, J., Liu, H. L., Solomon, S. C., Polvani, L. M., Emmons, L. K., Lamarque, J. F., Richter, J. H., Glanville, A. S., Bacmeister, J. T., Phillips, A. S., Neale, R. B.,

Simpson, I. R., DuVivier, A. K., Hodzic, A., and Randel, W. J.: The Whole Atmosphere Community Climate Model Version 6 (WACCM6), Journal of Geophysical Research: Atmospheres, n/a, 10.1029/2019JD030943, 2019.

Hauglustaine, D. A., Balkanski, Y., and Schulz, M.: A global model simulation of present and future nitrate aerosols and their direct radiative forcing of climate, Atmos. Chem. Phys., 14, 11031-11063, 10.5194/acp-14-11031-2014, 2014.

Horowitz, L. W., Walters, S., Mauzerall, D. L., Emmons, L. K., Rasch, P. J., Granier, C., Tie, X., Lamarque, J.-F., Schultz, M. G., Tyndall, G. S., Orlando, J. J., and Brasseur, G. P.: A global simulation of tropospheric ozone and related tracers: Description and evaluation of MOZART, version 2, Journal of Geophysical Research: Atmospheres, 108, 10.1029/2002jd002853, 2003.

Horowitz, L. W., Naik, V., Paulot, F., Ginoux, P. A., Dunne, J. P., Mao, J., Schnell, J., Chen, X., He, J., John, J. G., Lin, M., Lin, P., Malyshev, S., Paynter, D., Shevliakova, E., and Zhao, M.: The GFDL Global Atmospheric Chemistry-Climate Model AM4.1: Model Description and Simulation Characteristics, Journal of Advances in Modeling Earth Systems, n/a, e2019MS002032, 10.1029/2019ms002032, 2020.

Jones, A., Roberts, D. L., Woodage, M. J., and Johnson, C. E.: Indirect sulphate aerosol forcing in a climate model with an interactive sulphur cycle, Journal of Geophysical Research: Atmospheres, 106, 20293-20310, 10.1029/2000jd000089, 2001.

Khairoutdinov, M., and Kogan, Y.: A New Cloud Physics Parameterization in a Large-Eddy Simulation Model of Marine Stratocumulus, Monthly Weather Review, 128, 229-243, 10.1175/1520-0493(2000)128<0229:ancppi>2.0.co;2, 2000.

Kirkevåg, A., Grini, A., Olivié, D., Seland, Ø., Alterskjær, K., Hummel, M., Karset, I. H. H., Lewinschal, A., Liu, X., Makkonen, R., Bethke, I., Griesfeller, J., Schulz, M., and Iversen, T.: A production-tagged aerosol module for Earth system models, OsloAero5.3 – extensions and updates for CAM5.3-Oslo, Geosci. Model Dev., 11, 3945-3982, 10.5194/gmd-11-3945-2018, 2018.

Kuhlbrodt, T., Jones, C. G., Sellar, A., Storkey, D., Blockley, E., Stringer, M., Hill, R., Graham, T., Ridley, J., Blaker, A., Calvert, D., Copsey, D., Ellis, R., Hewitt, H., Hyder, P., Ineson, S., Mulcahy, J., Siahaan, A., and Walton, J.: The Low-Resolution Version of HadGEM3 GC3.1: Development and Evaluation for Global Climate, Journal of Advances in Modeling Earth Systems, 10, 2865-2888, 10.1029/2018ms001370, 2018.

Lurton, T., Balkanski, Y., Bastrikov, V., Bekki, S., Bopp, L., Braconnot, P., Brockmann, P., Cadule, P., Contoux, C., Cozic, A., Cugnet, D., Dufresne, J.-L., Éthé, C., Foujols, M.-A., Ghattas, J., Hauglustaine, D., Hu, R.-M., Kageyama, M., Khodri, M., Lebas, N., Levavasseur, G., Marchand, M., Ottlé, C., Peylin, P., Sima, A., Szopa, S., Thiéblemont, R., Vuichard, N., and Boucher, O.: Implementation of the CMIP6 Forcing Data in the IPSL-CM6A-LR Model, Journal of Advances in Modeling Earth Systems, 12, e2019MS001940, 10.1029/2019ms001940, 2020.

Mann, G. W., Carslaw, K. S., Spracklen, D. V., Ridley, D. A., Manktelow, P. T., Chipperfield, M. P., Pickering, S. J., and Johnson, C. E.: Description and evaluation of GLOMAP-mode: a modal global aerosol microphysics model for the UKCA composition-climate model, Geosci. Model Dev., 3, 519-551, 10.5194/gmd-3-519-2010, 2010.

[revised manuscript text omitted]

Walters, D., Baran, A. J., Boutle, I., Brooks, M., Earnshaw, P., Edwards, J., Furtado, K., Hill, P., Lock, A., Manners, J., Morcrette, C., Mulcahy, J., Sanchez, C., Smith, C., Stratton, R., Tennant, W., Tomassini, L., Van Weverberg, K., Vosper, S., Willett, M., Browse, J., Bushell, A., Carslaw, K.,

Dalvi, M., Essery, R., Gedney, N., Hardiman, S., Johnson, B., Johnson, C., Jones, A., Jones, C., Mann, G., Milton, S., Rumbold, H., Sellar, A., Ujiie, M., Whitall, M., Williams, K., and Zerroukat, M.: The Met Office Unified Model Global Atmosphere 7.0/7.1 and JULES Global Land 7.0 configurations, Geosci. Model Dev., 12, 1909-1963, 10.5194/gmd-12-1909-2019, 2019.

Wang, R., Balkanski, Y., Boucher, O., Ciais, P., Schuster, G. L., Chevallier, F., Samset, B. H., Liu, J., Piao, S., Valari, M., and Tao, S.: Estimation of global black carbon direct radiative forcing and its uncertainty constrained by observations, Journal of Geophysical Research: Atmospheres, 121, 5948-5971, 10.1002/2015jd024326, 2016.

Watanabe, M., Suzuki, T., O'ishi, R., Komuro, Y., Watanabe, S., Emori, S., Takemura, T., Chikira, M., Ogura, T., Sekiguchi, M., Takata, K., Yamazaki, D., Yokohata, T., Nozawa, T., Hasumi, H., Tatebe, H., and Kimoto, M.: Improved Climate Simulation by MIROC5: Mean States, Variability, and Climate Sensitivity, Journal of Climate, 23, 6312-6335, 10.1175/2010jcli3679.1, 2010.

Williams, K. D., Copsey, D., Blockley, E. W., Bodas-Salcedo, A., Calvert, D., Comer, R., Davis, P., Graham, T., Hewitt, H. T., Hill, R., Hyder, P., Ineson, S., Johns, T. C., Keen, A. B., Lee, R. W., Megann, A., Milton, S. F., Rae, J. G. L., Roberts, M. J., Scaife, A. A., Schiemann, R., Storkey, D., Thorpe, L., Watterson, I. G., Walters, D. N., West, A., Wood, R. A., Woollings, T., and Xavier, P. K.: The Met Office Global Coupled Model 3.0 and 3.1 (GC3.0 and GC3.1) Configurations, Journal of Advances in Modeling Earth Systems, 10, 357-380, 10.1002/2017ms001115, 2018.

Woodward, S.: Modeling the atmospheric life cycle and radiative impact of mineral dust in the Hadley Centre climate model, Journal of Geophysical Research: Atmospheres, 106, 18155-18166, 10.1029/2000jd900795, 2001.

Wu, T., Zhang, F., Zhang, J., Jie, W., Zhang, Y., Wu, F., Li, L., Liu, X., Lu, X., Zhang, L., Wang, J., and Hu, A.: Beijing Climate Center Earth System Model version 1 (BCC-ESM1): Model Description and Evaluation, Geosci. Model Dev. , 13, 977-1005, 10.5194/gmd-2019-172, 2020.

Yukimoto, S., Kawai, H., Koshiro, T., Oshima, N., Yoshida, K., Urakawa, S., Tsujino, H., Deushi, M., Tanaka, T., Hosaka, M., Yabu, S., Yoshimura, H., Shindo, E., Mizuta, R., Obata, A., Adachi, Y., and Ishii, M.: The Meteorological Research Institute Earth System Model Version 2.0, MRI-ESM2.0: Description and Basic Evaluation of the Physical Component, J. Meteor. Soc. Japan, 97, 931-965, 10.2151/jmsj.2019-051, 2019.

---

## Author Response (AR2)

Response to the editor's comments on **Effective radiative forcing from emissions of reactive gases and aerosols – a multi-model comparison, G. Thornhill et al.**

We appreciate the careful reading of the editor and would like to thank him for his comments and corrections. Our responses are in blue.
* * *
Comments to the Author:

L31-33: Please synthesize the description of qualified present-day ERFs consistent in these two sentences.
We have re-worded these sentences to make sure the definition of the ERF is consistent in the first two sentences.

L38-39: This sentence alone is not ideal to form a separate paragraph.
This has been incorporated into the preceding paragraph.

L40-42: This long sentence is ambiguous. Please revise.
This has been re-written to remove ambiguity.

L44: The "," should be replaced with "." for a complete sentence.
Corrected.

L103: "." is missing at the end of the sentence.
Corrected.

L220-221: please use a better way to describe or define these variables than using the equal sign. This has been re-written to clarify the terms without using the '=' sign.

Fig.1: Please clarify in the caption what types of aerosols "aer" includes. It's unclear why NH3-related ERF is placed at the right-most column.
The plot has been re-organised, and text added to clarify the constituents in 'aer'.

Fig. 2: There is a typo in y-axis title "Radiative Flux" for all panels. I understand that there are different numbers of participating models in the different experiments/panels; however, it would still be nice to vertically align the same individual models in all panels (i.e., with a total of 8 models but leaving out any missing models in panel a, b, and c). In this way, all four panels can be made consistent in font and bar sizes. Same for Fig. 6.
The figures have been re-done, so the typo has been corrected, and the inclusion of models without data for specific experiments has been included to allow for vertical alignment, and consistency in the panels font and bar sizes.
Table 3: the column title "cs,af" needs to be made consistent with the notation used in the next (ERFcs,af?). Same for Table 5.
This has been corrected so the table headings are consistent with the text.

Table 4: I don't see a column for IRFair/AOD that is described in the title.
The reference to the column has been removed from the title.

Fig. 9: the title "Methane lifetime" is misleading. Please remove or use a more accurate one.
The title has been changed to reflect the fact that the plot is for the change in methane lifetime.

[revised manuscript text omitted]